# MULTISIZE DATASET CONDENSATION

**Yang He**[1,2], **Lingao Xiao**[1,2], **Joey Tianyi Zhou**[1,2*], **Ivor W. Tsang**[1,2,3]
[1]CFAR, Agency for Science, Technology and Research, Singapore
[2]IHPC, Agency for Science, Technology and Research, Singapore
[3]School of Computer Science and Engineering, Nanyang Technological University
{He_Yang, Joey_Zhou, Ivor_Tsang}@cfar.a-star.edu.sg

## ABSTRACT

While dataset condensation effectively enhances training efficiency, its application in on-device scenarios brings unique challenges. 1) Due to the fluctuating computational resources of these devices, there's a demand for a flexible dataset size that diverges from a predefined size. 2) The limited computational power on devices often prevents additional condensation operations. These two challenges connect to the "subset degradation problem" in traditional dataset condensation: a subset from a larger condensed dataset is often unrepresentative compared to directly condensing the whole dataset to that smaller size. In this paper, we propose Multisize Dataset Condensation (MDC) by **compressing $N$ condensation processes into a single condensation process to obtain datasets with multiple sizes.** Specifically, we introduce an "adaptive subset loss" on top of the basic condensation loss to mitigate the "subset degradation problem". Our MDC method offers several benefits: 1) No additional condensation process is required; 2) reduced storage requirement by reusing condensed images. Experiments validate our findings on networks including ConvNet, ResNet and DenseNet, and datasets including SVHN, CIFAR-10, CIFAR-100 and ImageNet. For example, we achieved 5.22%-6.40% average accuracy gains on condensing CIFAR-10 to ten images per class. Code is available at: https://github.com/he-y/Multisize-Dataset-Condensation.

## 1 INTRODUCTION

With the explosive growth in data volume, dataset condensation has emerged as a crucial tool in deep learning, allowing models to train more efficiently by focusing on a reduced set of informative data points. However, data processing faces new challenges as more applications transition to on-device processing (Cai et al., 2020; Lin et al., 2022; Yang et al., 2022; 2023; Qiu et al., 2022; Lee & Yoo, 2021; Dhar et al., 2021), whether due to security concerns, real-time demands, or connectivity issues. Such devices' inherently fluctuating computational resources require flexible dataset sizes, deviating from the conventional condensed datasets. However, this request for flexibility surfaces a critical concern since the additional condensation process is unfeasible on these resource-restricted devices.

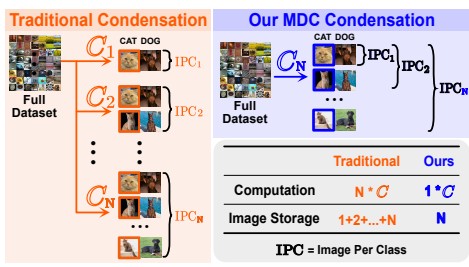

Figure 1: Condense datasets to multiple sizes requires $N$ separate traditional condensation processes (left) but just a **single** MDC processes (right).

Why not select a **subset** from a condensed dataset for on-device scenarios? We find the "subset degradation problem" in traditional dataset condensation: if we select a subset from a condensed dataset, the performance of the subset is much lower than directly condensing the full dataset to the target small size. An intuitive solution would be to conduct the condensation process $N$ times. However, since each process requires 200K epochs and these processes cumulatively generate $1 + 2 + \ldots + N$ images (left figure of Fig. 1), it is not practical for on-device applications.

---

*Corresponding Author.
    Lingao Xiao is an intern student under the supervision of Yang He.

To address these issues, we present the Multisize Dataset Condensation (MDC) method to compress $N$ condensation processes into just one condensation process, resulting in just one dataset (right figure of Fig. 1). We propose the novel **"adaptive subset loss"** on top of the "base loss" in the condensation process to alleviate the "subset degradation problem" for all subsets. **"Adaptive"** means we adaptively select the Most Learnable Subset (MLS) from $N - 1$ subsets for different condensation iterations. The **"subset loss"** refers to the loss computed from the chosen MLS, which is then utilized to update the corresponding subset.

How to select the Most Learnable Subset (**MLS**)? We integrate this selection process into the traditional condensation process with two loops: an outer loop for weight initialization and an inner loop for network training. MLS selection has three components. (i) **Feature Distance Calculation**: Evaluating distances between all subsets and the real dataset, where smaller distances suggest better representation. For each outer loop iteration, we calculate the average Feature Distance over all the inner training epochs. (ii) **Feature Distance Comparison**: We compare the average feature distances at two outer loops. A large rate of change in distances denotes the current subset has high learning potential and should be treated as MLS. (iii) **MLS Freezing Judgement**: To further mitigate the "subset degradation problem", our updating strategy depends on the MLS' size relative to its predecessor. If the recent MLS exceeds its predecessor in size, we freeze the older MLS and only update its non-overlapping elements in the newer MLS. Otherwise, we update the entire newer MLS.

The key contributions of our work are: **1)** To the best of our knowledge, it's the first work to condense the $N$ condensation processes into a single condensation process. **2)** We firstly point out the "subset degradation problem" and propose "adaptive subset loss" to mitigate the problem. **3)** Our method is validated with extensive experiments on networks including ConvNet, ResNet and DenseNet, and datasets including SVHN, CIFAR-10, CIFAR-100 and ImageNet.

## 2 RELATED WORKS

**Matching Objectives.** The concept of dataset condensation, or distillation, is brought up by Wang et al. (2018). The aim is to learn a synthetic dataset that is equally effective but much smaller in size. **1) Gradient Matching** (Zhao et al., 2021; Jiang et al., 2022; Lee et al., 2022b; Loo et al., 2023) methods propose to match the network gradients computed by the real dataset and the synthetic dataset. **2) Other matching objectives** include performance matching (Wang et al., 2018; Nguyen et al., 2021a;b; Zhou et al., 2022; Loo et al., 2022), distribution or feature matching (Zhao & Bilen, 2023; Wang et al., 2022; Zhao et al., 2023), trajectory matching (Cazenavette et al., 2022; Du et al., 2023; Cui et al., 2023), representative matching (Liu et al., 2023b; Tukan et al., 2023), loss-curvature matching (Shin et al., 2023), and BN matching (Yin et al., 2023; Yin & Shen, 2023). However, all the aforementioned methods suffer from the "subset degradation problem," failing to provide a solution.

**Better Optimization.** Various methods are proposed to improve the condensation process, including data augmentation (Zhao & Bilen, 2021), data parameterization (Deng & Russakovsky, 2022; Liu et al., 2022; Kim et al., 2022b; Nooralinejad et al., 2022; Kim et al., 2022a; Sun et al., 2023), model augmentation (Zhang et al., 2023b), and model pruning (Li et al., 2023). Our method can combine with these methods to achieve better performance.

**Condensation with GANs.** Several works (Zhao & Bilen, 2022; Lee et al., 2022a; Cazenavette et al., 2023) leverage Generative Adversarial Networks (GANs) to enhance the condensation process. For instance, Wang et al. (2023) generate images by feeding noise into a trained generator, whereas Zhang et al. (2023a) employ learned codebooks for synthesis. The drawback is that they require substantially more storage to save the model and demand 20% more computational power during deployment (Wang et al., 2023). In contrast, our solution delivers immediately usable condensed images, ensuring efficiency in both storage and computation.

**Comparison with Slimmable Dataset Condensation (SDC; Liu et al. (2023a)).** SDC aims to extract a smaller synthetic dataset given the previous condensation results. The differences include: 1) SDC needs two separate condensation processes, while our method just needs one; 2) SDC relies on the condensed dataset, but our method does not; 3) SDC requires computational-intensive singular value decomposition (SVD) on condensed images, while our condensed images can be directly used for application.

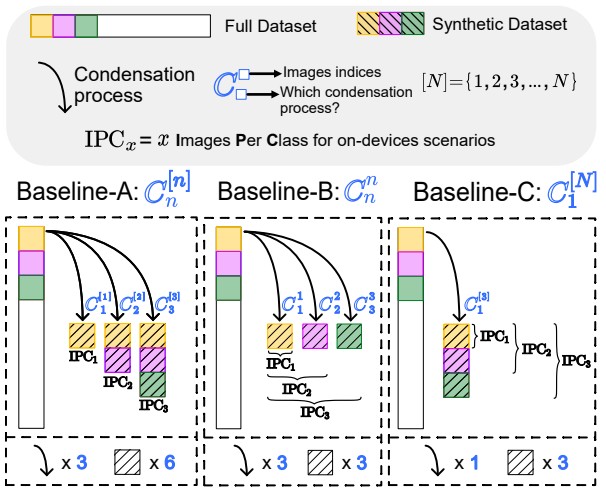

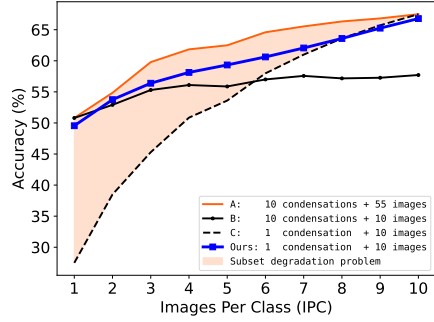

| | Symbol | Condense | Storage |
|---|---|---|---|
| A | $\mathbb{C}_n^{[n]}, n \in \{1, 2, \ldots, N\}$ | $N$ | $1 + 2 + \ldots + N$ |
| B | $\mathbb{C}_n^n, n \in \{1, 2, \ldots, N\}$ | $N$ | $N$ |
| C | $\mathbb{C}_1^{[N]}$ | 1 | $N$ |
| Ours | $\mathbb{C}_1^{[N]}$ | 1 | $N$ |

(b) Resources required for different baselines. $[n]$ represents the set of $\{1, 2, \ldots, n\}$.

(a) "Multi-size condensation" with three baselines to obtain condensed datasets of size 1,2,3. **Left:** Baseline A conducts 3 separate condensation processes and stores $1 + 2 + 3 = 6$ images. **Middle:** Baseline B performs 3 times IPC$_1$ condensation with different image indices as initialization and in total stores 3 images. **Right:** Baseline C condenses once and stores 3 images. Multiple sizes are achieved with subset selection.

(c) On CIFAR-10, the accuracy of three proposed baselines for "**multi-size condensation**" to get IPC size from 1 to 10.

Figure 2: Three different baselines for multi-size condensation.

## 3 METHOD

### 3.1 PRELIMINARIES

Given a big original dataset $\mathcal{B} \in \mathbb{R}^{M \times d}$ with $M$ number of $d$-dimensional data, the objective is to obtain a small synthetic dataset $\mathcal{S} \in \mathbb{R}^{N \times d}$ where $N \ll M$. Leveraging the gradient-based technique (Zhao et al., 2021), we minimize the gradient distance between big dataset $\mathcal{B}$ and synthetic dataset $\mathcal{S}$:

$$\min_{\mathcal{S} \in \mathbb{R}^{N \times d}} D\left(\nabla_\theta \ell(\mathcal{S}; \theta), \nabla_\theta \ell(\mathcal{B}; \theta)\right) = D(\mathcal{S}, \mathcal{B}; \theta), \tag{1}$$

where the function $D(\cdot)$ is defined as a distance metric such as MSE, $\theta$ represents the model parameters, and $\nabla_\theta \ell(\cdot)$ denotes the gradient, utilizing either the big dataset $\mathcal{B}$ or its synthetic version $\mathcal{S}$. During condensation, the synthetic dataset $\mathcal{S}$ and model $\theta$ are updated alternatively,

$$\mathcal{S} \leftarrow \mathcal{S} - \lambda \nabla_\mathcal{S} D(\mathcal{S}, \mathcal{B}; \theta), \quad \theta \leftarrow \theta - \eta \nabla_\theta \ell(\theta; \mathcal{S}), \tag{2}$$

where $\lambda$ and $\eta$ are learning rates designated for $\mathcal{S}$ and $\theta$, respectively.

### 3.2 SUBSET DEGRADATION PROBLEM

To explain the "subset degradation problem", we name the condensation process in Eq. 1 and Eq. 2 as "**basic condensation**". This can be symbolized by $\mathbb{C}_1^{[N]}$, where $[N] = \{1, 2, 3, \ldots, N\}$. The subscript of $\mathbb{C}_\square^\square$ indicates the index of the condensation process, while the superscript of $\mathbb{C}_\square^\square$ represents the index of original images that are used as the initialization for condensation.

For on-device applications, we need condensed datasets with multiple sizes, namely, "**multi-size condensation**". Inspired by He et al. (2024); Yin et al. (2023), we introduce three distinct baselines for "multi-size condensation": Baseline-A, Baseline-B, and Baseline-C, denoted as $\mathbb{C}_n^{[n]}$, $\mathbb{C}_n^n$, and $\mathbb{C}_1^{[N]}$, respectively. Fig. 2a illustrates how to conduct "multi-size condensation" to obtain the condensed dataset with sizes 1, 2, and 3 with our proposed baselines. Baseline-A employs **three** basic condensations of varying sizes, yielding **six** images. Baseline-B uses **three** size-1 basic condensations

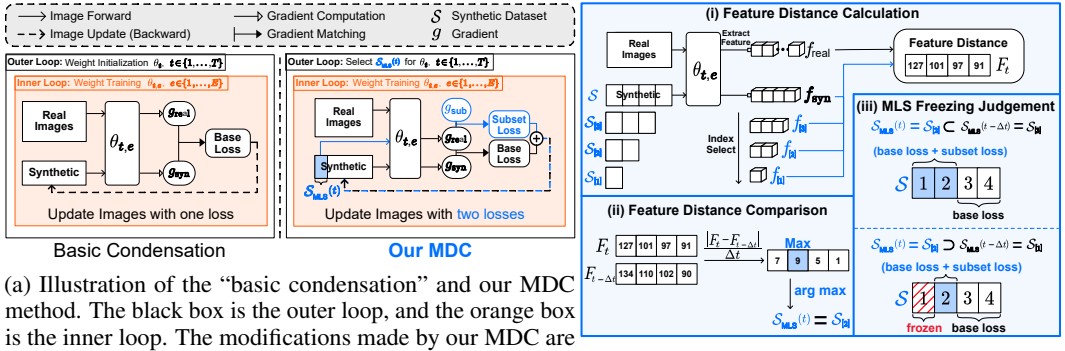

(a) Illustration of the "basic condensation" and our MDC method. The black box is the outer loop, and the orange box is the inner loop. The modifications made by our MDC are marked with blue.

(b) Illustration of MLS selection process.

Figure 3: Explanation of Our MDC.

but varies by image index, resulting in **three** unique images. Baseline-C adopts a **single** basic condensation to get **three** images, subsequently selecting image subsets for flexibility. Fig. 2b presents the count of required basic condensation processes and the storage demands in terms of image numbers.

In Fig. 2c, we highlight the "subset degradation problem" using Baseline-A's orange line and Baseline-C's black dashed line. Baseline-A requires ten condensation processes, while Baseline-C just condenses once and selects **subsets** of the condensed dataset sized at 10. The shaded orange region indicates a notable accuracy drop for the subset when compared to the basic-condensed dataset. A critical observation is that the accuracy discrepancy grows as the subset's size becomes smaller.

### 3.3 MULTISIZE DATASET CONDENSATION

#### 3.3.1 SUBSET LOSS TO COMPRESS CONDENSATION PROCESSES

To address the "subset degradation problem", the objective function now becomes:

$$\min_{\mathcal{S}\in\mathbb{R}^{N\times d}} D\left(\nabla_\theta \ell\left(\mathcal{S}_{[1]}, \mathcal{S}_{[2]}, \ldots \mathcal{S}_{[N]}; \theta\right), \nabla_\theta \ell\left(\mathcal{B}; \theta\right)\right), \tag{3}$$

where $\mathcal{S}_{[n]} = \mathcal{S}_{\{1,2,\ldots,n\}} \subset \mathcal{S} = \mathcal{S}_{[N]}$ represents $n_{th}$ subset of the synthetic dataset $\mathcal{S} \in \mathbb{R}^{N\times d}$. We want each subset $\mathcal{S}_{[n]}$ to have a small distance from the big dataset $\mathcal{B}$. $\mathcal{S}_{[N]}$ contributes the "base loss", and $\mathcal{S}_{[1],[2],\ldots,[N-1]}$ contribute to the "subset loss". Note that the subsets also have a relationship with each other. For instance, the $2_{nd}$ subset $\mathcal{S}_{[2]} = \mathcal{S}_{\{1,2\}}$ is also a subset of the $4_{th}$ subset $\mathcal{S}_{[4]} = \mathcal{S}_{\{1,2,3,4\}}$.

We aim to incorporate the information of subsets without requiring additional condensation processes or extra images. To achieve this, we need to **compress** the information from the $N-1$ different **condensation processes** of Baseline-A, including $\mathbb{C}_1^{[1]}, \mathbb{C}_2^{[2]}, \ldots, \mathbb{C}_{N-1}^{[N-1]}$, into the process $\mathbb{C}_N^{[N]}$. We propose the "subset loss" on top of the "base loss" to achieve this purpose in a single condensation process. The "base loss" is used to maintain the basic condensation process $\mathbb{C}_N^{[N]}$, while the "subset loss" is used to enhance the learning process of the subsets via $\mathbb{C}_1^{[1]}, \mathbb{C}_2^{[2]}, \ldots, \mathbb{C}_{N-1}^{[N-1]}$. We have a new updating strategy:

$$\mathcal{S} \leftarrow \mathcal{S} - \lambda\left(\nabla_{\mathcal{S}} D\left(\mathcal{S}, \mathcal{B}; \theta\right) + \nabla_{\mathcal{S}_{[n]}} D\left(\mathcal{S}_{[n]}, \mathcal{B}; \theta\right)\right), \quad n \in [1, N-1]. \tag{4}$$

where $\mathcal{S} = \mathcal{S}_{[N]}$ represents the condensed dataset of $N$ images and is associated with the "base loss". $\mathcal{S}_{[n]}$ is subset and contributes to the "subset loss". A comparison between Eq. 4 and basic condensation is shown in Fig. 3a. As depicted in Fig. 2b, our technique aligns with Baseline-C in terms of the counts of both condensation processes and images.

#### 3.3.2 SELECTING MLS FOR ADAPTIVE SUBSET LOSS

Among the $N-1$ subsets from $\{\mathcal{S}_{[1]}, \mathcal{S}_{[2]}, \ldots, \mathcal{S}_{[N-1]}\}$, we identify a particularly representative subset, $\mathcal{S}[n^*]$, where $n^* \in [1, N-1]$. We term this the Most Learnable Subset (MLS). In each condensation iteration, the MLS is selected **adaptively** to fit that particular iteration. Our approach

relies on three components to determine the MLS. Each component is illustrated in Fig. 3b. The algorithm of the proposed method is shown in Appendix A.

**Feature Distance Calculation (Fig. 3b-(i)).** Eq. 3 represents the traditional approach for computing the gradient distance between subsets $\{\mathcal{S}_{[1]}, \mathcal{S}_{[2]}, \ldots \mathcal{S}_{[N-1]}\}$ and the big dataset $\mathcal{B}$. This method requires gradient calculations to be performed $N-1$ times across the $N-1$ subsets, leading to considerable computational overhead. To alleviate this, we introduce the concept of "feature distance" as a substitute for the "gradient distance" to reduce computation while capturing essential characteristics among subsets. The feature distance at a specific condensation iteration $t$ for subset $\mathcal{S}_{[n]}$ can be represented as:

$$F_t\left(\mathcal{S}_{[n]}, \mathcal{B}\right) = D\left(f_t\left(\mathcal{S}_{[n]}\right), f_t(\mathcal{B})\right), \tag{5}$$

where $f_t(\cdot)$ is the feature extraction function for $t_{th}$ condensation iteration, and $D(\cdot)$ is a distance metric like MSE. For subsets, the gradient distance mandates $N-1$ forward passes and an equal number of backward passes for a total of $N-1$ subsets. In contrast, the feature distance requires only a single forward pass and no backward pass. This is because the features are hierarchically arranged, and the feature set derived from a subset of size $n$ can be straightforwardly extracted from the features of the larger dataset of size $N$.

**Feature Distance Comparison (Fig. 3b-(ii)).** Generally, as the size of the subset increases, the feature distance diminishes. This is because the larger subset is more similar to the big dataset $\mathcal{B}$. Let's consider two subsets $\mathcal{S}_{[p]}$ and $\mathcal{S}_{[q]}$ such that $1 < p < q < N$. This implies that the size of subset $\mathcal{S}_{[p]}$ is less than the size of subset $\mathcal{S}_{[q]}$. Their feature distances at iteration $t$ can be represented as:

$$F_t\left(\mathcal{S}_{[p]}, \mathcal{B}\right) > F_t\left(\mathcal{S}_{[q]}, \mathcal{B}\right), \quad \text{if} \ \ 1 < p < q < N. \tag{6}$$

Initially, it is intuitive that $\mathcal{S}_{[1]}$, being the smallest subset, would manifest the greatest distance or disparity when compared to $\mathcal{B}$. As such, $\mathcal{S}_{[1]}$ should be the MLS at the beginning of the condensation process. As the condensation process progresses, we have:

$$\underbrace{F_{t-\Delta t}\left(\mathcal{S}_{[p]}, \mathcal{B}\right) > F_t\left(\mathcal{S}_{[p]}, \mathcal{B}\right)}_{p}, \quad \underbrace{F_{t-\Delta t}\left(\mathcal{S}_{[q]}, \mathcal{B}\right) > F_t\left(\mathcal{S}_{[q]}, \mathcal{B}\right)}_{q}, \tag{7}$$

where $t - \Delta t$ and $t$ are two different time points for the condensation process. The reason for $F_{t-\Delta t} > F_t$ is that the subsets get more representative as the condensation progresses, causing their feature distances to shrink. So, the most **learnable** subset would be the one whose feature distance reduction rate is the **highest**. The feature distance reduction rate is:

$$R(\mathcal{S}_{[n]}, t) = \frac{\Delta F_{\mathcal{S}_{[n]}}}{\Delta t} = \frac{\left|F_t\left(\mathcal{S}_{[n]}, \mathcal{B}\right) - F_{t-\Delta t}\left(\mathcal{S}_{[n]}, \mathcal{B}\right)\right|}{\Delta t}, \tag{8}$$

where $R(\mathcal{S}_{[n]}, t)$ represents the rate of change of feature distance for subset $\mathcal{S}_{[n]}$ at the time point $t$, and $\Delta F_{\mathcal{S}_{[n]}}$ denotes the change in feature distance of subset $\mathcal{S}_{[n]}$ from time $t - \Delta t$ to $t$. An example for feature distance calculation can be found in Appendix B.2. The MLS for the time $t$ can be described as:

$$\mathcal{S}_{\text{MLS}}(t) = \mathcal{S}_{[n_t^*]} = \underset{\mathcal{S}_{[n]}}{\arg\max}\left(R\left(\mathcal{S}_{[n]}, t\right)\right) \quad \text{where} \ \ n \in [1, N-1]. \tag{9}$$

Eq. 9 seeks the subset that has the steepest incline or decline in its feature distance from $\mathcal{B}$ over the time interval $\Delta t$. This indicates the subset is "learning" at the **fastest rate**, thus deeming it the most learnable subset (MLS).

**MLS Freezing Judgement (Fig. 3b-(iii)).** To further reduce the impact of the "subset degradation problem", we modify the updating strategy in Eq. 4. The judgement will be modified if the current MLS differs in size from its predecessor; otherwise, it remains unchanged:

$$\text{Using Eq. 4 to} \begin{cases} \text{Update} \ \ \mathcal{S} & \text{if } \mathcal{S}_{\text{MLS}}(t) \subset \mathcal{S}_{\text{MLS}}(t - \Delta t) \\ \text{Update} \ \ \mathcal{S} \setminus \mathcal{S}_{\text{MLS}}(t - \Delta t) & \text{if } \mathcal{S}_{\text{MLS}}(t) \supset \mathcal{S}_{\text{MLS}}(t - \Delta t) \end{cases} \tag{10}$$

where $\setminus$ is the symbol for set minus. If the size of the current MLS $\mathcal{S}_{\text{MLS}}(t)$ is smaller than its predecessor $\mathcal{S}_{\text{MLS}}(t - \Delta t)$, we update the entire synthetic data $\mathcal{S}$ with Eq. 4. However, when the size of the current MLS is larger than its predecessor, updating the entire $\mathcal{S}$ would cause the optimized predecessor to be negatively affected by new gradients. Therefore, we freeze the preceding MLS $\mathcal{S}_{\text{MLS}}(t - \Delta t)$ to preserve already learned information, as shown in the red shadowed $\mathcal{S}_{[1]}$ in Fig. 3b-(iii). As a result, only the non-overlapping elements, i.e., $\mathcal{S} \setminus \mathcal{S}_{\text{MLS}}(t - \Delta t)$ are updated.

| Dataset | | | 1 | 2 | 3 | 4 | 5 | 6 | 7 | 8 | 9 | 10 | Avg. | Diff. |
|---|---|---|---|---|---|---|---|---|---|---|---|---|---|---|
| SVHN | | A | 68.50† | 75.27 | 79.55 | 81.85 | 83.33 | 84.53 | 85.66 | 86.05 | 86.25 | 87.50† | 81.85 | - |
| | | B | 68.50† | 71.65 | 71.27 | 71.92 | 73.28 | 70.74 | 71.83 | 71.08 | 71.97 | 71.55 | 71.38 | - |
| | | C | 35.48 | 51.55 | 60.42 | 67.97 | 74.38 | 77.65 | 81.70 | 83.86 | 85.96 | 87.50† | 70.65 | 0 |
| | | Ours | 63.26 | 67.91 | 72.15 | 74.09 | 77.54 | 78.17 | 80.92 | 82.82 | 84.27 | 86.38 | 76.75 | +6.10 |
| CIFAR-10 | | A | 50.80 | 54.85 | 59.79 | 61.84 | 62.49 | 64.59 | 65.53 | 66.33 | 66.82 | 67.50† | 62.05 | - |
| | | B | 50.80 | 53.17 | 55.09 | 56.17 | 55.80 | 56.98 | 57.60 | 57.78 | 58.22 | 58.38 | 56.00 | - |
| | | C | 27.49 | 38.50 | 45.29 | 50.85 | 53.60 | 57.98 | 60.99 | 63.60 | 65.71 | 67.50† | 53.15 | 0 |
| | | Ours | 49.66 | 54.58 | 53.92 | 54.55 | 55.18 | 58.80 | 61.51 | 63.36 | 65.41 | 66.72 | 58.37 | +5.22 |
| CIFAR-100 | | A | 28.90† | 34.28 | 37.35 | 39.13 | 41.15 | 42.65 | 43.62 | 44.48 | 45.07 | 45.40 | 40.20 | - |
| | | B | 28.90† | 30.63 | 31.64 | 31.76 | 32.61 | 32.85 | 33.03 | 33.04 | 33.32 | 33.39 | 32.12 | - |
| | | C | 14.38 | 21.76 | 28.01 | 32.21 | 35.27 | 39.09 | 40.92 | 42.69 | 44.28 | 45.40 | 34.40 | 0 |
| | | Ours | 27.58 | 31.83 | 33.59 | 35.42 | 36.93 | 38.95 | 40.70 | 42.05 | 43.86 | 44.34 | 37.53 | +3.13 |

(a) Results of SVHN, CIFAR-10, CIFAR-100 targeting IPC$_{10}$.

| Dataset | | | 1 | 2 | 3 | 4 | 5 | 6 | 7 | 8 | 9 | 10 | 20 | 30 | 40 | 50 | Avg. | Diff. |
|---|---|---|---|---|---|---|---|---|---|---|---|---|---|---|---|---|---|---|
| SVHN | | A | 68.50† | 75.27 | 79.55 | 81.85 | 83.33 | 84.53 | 85.66 | 86.05 | 86.25 | 87.50† | 89.54 | 90.27 | 91.09 | 91.38 | 84.34 | - |
| | | C | 34.90 | 46.52 | 52.23 | 56.30 | 62.25 | 65.34 | 68.84 | 69.57 | 71.95 | 74.69 | 83.73 | 87.83 | 89.73 | 91.38 | 68.23 | 0 |
| | | Ours | 58.77 | 67.72 | 69.33 | 72.26 | 75.02 | 73.71 | 74.50 | 74.63 | 76.21 | 76.87 | 83.67 | 87.08 | 89.46 | 91.39 | 76.47 | +8.24 |
| CIFAR-10 | | A | 50.80 | 54.85 | 59.79 | 61.84 | 62.49 | 64.59 | 65.53 | 66.33 | 66.82 | 67.50† | 70.82 | 72.86 | 74.30 | 75.07 | 65.26 | - |
| | | C | 27.87 | 35.69 | 41.93 | 45.29 | 47.54 | 51.96 | 53.51 | 55.59 | 56.62 | 58.26 | 66.77 | 70.50 | 72.98 | 74.50 | 54.21 | 0 |
| | | Ours | 47.83 | 52.18 | 56.29 | 58.52 | 58.75 | 60.67 | 61.90 | 62.74 | 62.32 | 62.64 | 66.88 | 70.02 | 72.91 | 74.56 | 62.01 | +7.80 |
| CIFAR-100 | | A | 28.90 | 34.28 | 37.35 | 39.13 | 41.15 | 42.65 | 43.62 | 44.48 | 45.07 | 45.40 | 49.50 | 52.28 | 52.54 | 53.47 | 43.56 | - |
| | | C | 12.66 | 18.35 | 23.76 | 26.92 | 29.12 | 32.23 | 34.21 | 35.71 | 37.18 | 38.25 | 45.67 | 49.60 | 52.36 | 53.47 | 34.96 | 0 |
| | | Ours | 26.34 | 29.71 | 31.74 | 32.95 | 34.49 | 36.36 | 38.49 | 39.59 | 40.43 | 41.35 | 46.06 | 49.40 | 51.72 | 53.67 | 39.45 | +4.49 |

(b) Results of SVHN, CIFAR-10, CIFAR-100 targeting IPC$_{50}$.

| Dataset | | | 1 | 2 | 3 | 4 | 5 | 6 | 7 | 8 | 9 | 10 | 15 | 20 | Avg. | Diff. |
|---|---|---|---|---|---|---|---|---|---|---|---|---|---|---|---|---|
| ImageNet-10 | | A | 60.40 | 63.87 | 67.40 | 68.80 | 71.33 | 70.60 | 70.47 | 71.93 | 72.87 | 72.80† | 75.50 | 76.60† | 70.21 | - |
| | | B | 60.40 | 62.07 | 62.80 | 63.40 | 64.67 | 63.13 | 62.67 | 63.60 | 64.13 | 63.60 | 62.73 | 64.13 | 63.11 | - |
| | | C | 44.00 | 57.27 | 62.80 | 66.13 | 64.33 | 69.47 | 69.53 | 70.53 | 71.73 | 73.00 | 74.47 | 75.73 | 66.58 | 0 |
| | | Ours | 55.87 | 61.60 | 63.40 | 64.40 | 63.80 | 67.73 | 67.13 | 70.07 | 71.07 | 71.13 | 76.00 | 79.20 | 67.62 | +1.04 |

(c) Results of ImageNet-10 targeting IPC$_{20}$.

Table 1: Comparisons with three different baselines built with the IDC (Kim et al., 2022b). † denotes directly cited from original papers. Numbers with standard deviation can be found in Appendix B.4.

## 4 EXPERIMENTS

### 4.1 EXPERIMENT SETTINGS.

**Terms.** IPC$_n$ represents $n$ **I**mages **P**er **C**lass for the condensed dataset.

**Basic Condensation Training.** We use IDC (Kim et al., 2022b) to condense the CIFAR-10, CIFAR-100 (Krizhevsky et al., 2009) and SVHN (Netzer et al., 2011) with ConvNet-D3 (Gidaris & Komodakis, 2018). ImageNet-10 (Deng et al., 2009) is condensed via ResNet10-AP (He et al., 2016). For CIFAR-10, CIFAR-100 and SVHN, we use a batch size of 128 for IPC $\leq 30$, and a batch size of 256 for IPC $> 30$. For ImageNet-10 IPC$_{20}$, we use a batch size of 256. The network is randomly initialized 2000 times for CIFAR-10, CIFAR-100 and SVHN, and 500 times for ImageNet-10; for each initialization, the network is trained for 100 epochs. More details are provided in Appendix B.1.

**Basic Condensation Evaluations.** We also follow IDC (Kim et al., 2022b). For both ConvNet-D3 and ResNet10-AP, the learning rate is 0.01 with 0.9 momentum and 0.0005 weight decay. The SGD optimizer and a multi-step learning rate scheduler are used. The network is trained for 1000 epochs.

**MDC Settings. i) Feature Distance Calculation.** The last layer feature is used for the feature distance calculation. The computed feature distance is averaged across 100 inner loop training epochs for a specific outer loop. **ii) Feature Distance Comparison.** For CIFAR-10, CIFAR-100 and

|   | DC | DSA | MTT | IDC | DREAM | Ours |
|---|---|---|---|---|---|---|
| 1 | 15.35 | 16.76 | 18.80 | 27.49 | 32.52 | 49.66 |
| 2 | 19.75 | 21.22 | 24.90 | 38.50 | 39.57 | 54.58 |
| 3 | 22.54 | 26.78 | 31.90 | 45.29 | 48.21 | 53.92 |
| 4 | 26.28 | 30.18 | 38.10 | 50.85 | 53.84 | 54.55 |
| 5 | 30.37 | 33.43 | 43.20 | 53.60 | 55.25 | 55.18 |
| 6 | 33.99 | 38.15 | 49.20 | 57.98 | 60.46 | 58.80 |
| 7 | 36.36 | 41.18 | 51.60 | 60.99 | 63.27 | 61.51 |
| 8 | 39.83 | 45.37 | 56.30 | 63.60 | 65.04 | 63.36 |
| 9 | 42.68 | 49.21 | 58.50 | 65.71 | 67.40 | 65.41 |
| 10 | 44.90† | 52.10† | 62.80† | 67.50† | 69.40† | 66.72 |
| Avg. | 31.21 | 35.44 | 43.53 | 53.15 | 55.50 | **58.37** |
| Diff. | -27.16 | -22.93 | -14.84 | -5.22 | -2.87 | - |

(a) CIFAR-10, $IPC_{10}$.

|   | $\mathbf{R}$[1] | 1 | 2 | 5 | 10 | 20 | Avg. |
|---|---|---|---|---|---|---|---|
| LFS | (1, 20) | 20.34 | 23.69 | 28.58 | 35.39 | 42.47 | 30.09 |
| LBS | (20, 210) | 26.04 | 29.27 | 33.49 | 36.23 | 42.47 | 33.50 |
| Ours | (1, 20) | **27.66** | **31.09** | **35.50** | **41.56** | **49.30** | **37.02** |

(c) Comparing results with LFS and LBS (Liu et al., 2023a). CIFAR-100, $IPC_{20}$.

|   | DC | DSA | MTT | IDC | DREAM | Ours |
|---|---|---|---|---|---|---|
| 1 | 16.32 | 12.50 | 15.13 | 27.87 | 27.57 | 47.83 |
| 2 | 18.77 | 15.19 | 23.92 | 35.69 | 36.57 | 52.18 |
| 3 | 21.24 | 19.69 | 26.53 | 41.93 | 43.50 | 56.29 |
| 4 | 21.42 | 22.02 | 30.30 | 45.29 | 47.35 | 58.52 |
| 5 | 23.32 | 23.28 | 32.71 | 47.54 | 49.81 | 58.75 |
| 6 | 23.63 | 24.79 | 35.54 | 51.96 | 53.38 | 60.67 |
| 7 | 25.35 | 25.62 | 34.12 | 53.51 | 54.58 | 61.90 |
| 8 | 27.40 | 27.84 | 40.60 | 55.59 | 56.78 | 62.74 |
| 9 | 27.93 | 29.57 | 43.43 | 56.62 | 58.91 | 62.32 |
| 10 | 28.00 | 32.51 | 45.99 | 58.26 | 60.10 | 62.64 |
| 20 | 36.53 | 40.94 | 60.41 | 66.77 | 68.07 | 66.88 |
| 30 | 42.82 | 48.05 | 67.68 | 70.50 | 70.48 | 70.02 |
| 40 | 48.90 | 54.24 | 69.71 | 72.98 | 72.79 | 72.91 |
| 50 | 53.90† | 60.60† | 71.60† | 74.50† | 74.80† | 74.56 |
| Avg. | 29.68 | 31.20 | 42.69 | 54.21 | 55.33 | **62.01** |
| Diff. | -32.33 | -30.81 | -19.32 | -7.80 | -6.68 | - |

(b) CIFAR-10, $IPC_{50}$

Table 2: Comparison with SOTA condensation methods. † denotes directly cited from original papers.

SVHN, the feature distance is calculated at intervals of every $\Delta t = 100$ outer loop. For ImageNet-10, $\Delta t = 50$. **iii) MLS Freezing Judgement.** We follow Eq. 10 for MLS freezing.

## 4.2 PRIMARY RESULTS

**Comparison with Baseline-A, B, C.** Three baselines defined in Sec. 3.2, including Baseline-A,B,C, are created with IDC (Kim et al., 2022b). Tab. 1a and Tab. 1b provide the comparisons on three datasets: SVHN, CIFAR-10, and CIFAR-100 targeting $IPC_{10}$ and $IPC_{50}$; Tab. 1c provides the results on the ImageNet-10 dataset of $IPC_{20}$. As detailed in Fig. 2b, Baseline-C aligns with our condensation and storage requirements, so we mainly compare with Baseline-C. The results of Baseline-C are shaded in grey, while our method's accuracy is shaded in blue. Evidently, our approach consistently outperforms Baseline-C. For instance, on CIFAR-10 targeting $IPC_{10}$, our method improves 5.22% in average accuracy. The proposed method effectively addresses the "subset degradation problem" at small subsets. For $\mathcal{S}_{[1]}$ of $IPC_{10}$, we improve accuracy by +27.78% on SVHN, +22.17% on CIFAR-10, and +13.20% on CIFAR-100. Even though Baseline-B requires much more image storage ($N$ v.s. 1), our method beats Baseline-B for $IPC_{10}$ by +5.37% on SVHN, +2.37% on CIFAR-10, and +5.41% on CIFAR-100. The visualization of accuracies is presented in Fig. 2c.

**Comparison with State-of-the-art Methods.** In Tab. 2, we evaluate our approach against state-of-the-art (SOTA) condensation techniques including DC (Zhao et al., 2021), DSA (Zhao & Bilen, 2021), MTT (Cazenavette et al., 2022), IDC (Kim et al., 2022b), and DREAM (Liu et al., 2023b). From the table, it becomes evident that not only IDC (Kim et al., 2022b) but all condensation methods face the "subset degradation problem". Our MDC shows a clear advantage over other methods with a single condensation process and $IPC_N$ storage. Importantly, the accuracy of $\mathcal{S}_{[1]}$ is improved by 17.14% for $IPC_{10}$ and 20.26% for $IPC_{50}$ compared to DREAM (Liu et al., 2023b). Tab. 2c illustrates that our approach outperforms Slimmable DC (Liu et al., 2023a), another method for dataset flexibility. Notably, our method excels by 3.52% even against the resource-intensive LBS.

---

[1]$\mathbf{R}$ = (1, 20) means requiring one condensation process and storing 20 images.

| Calculate | Compare | Freeze | 1 | 2 | 3 | 4 | 5 | 6 | 7 | 8 | 9 | 10 | Avg. |
|---|---|---|---|---|---|---|---|---|---|---|---|---|---|
| - | - | - | 27.49 | 38.50 | 45.29 | 50.85 | 53.60 | 57.98 | 60.99 | 63.60 | 65.71 | 67.50 | 53.15 |
| ✓ | - | - | 49.35 | 48.27 | 50.00 | 52.30 | 54.20 | 58.29 | 60.90 | **63.63** | **65.90** | **67.63** | 57.08 |
| ✓ | ✓ | - | 40.12 | **54.91** | **56.02** | **56.12** | **56.18** | **59.74** | **61.68** | 63.41 | 65.56 | 67.01 | 58.08 |
| ✓ | ✓ | ✓ | **49.66** | 54.58 | 53.92 | 54.55 | 55.18 | 58.80 | 61.51 | 63.36 | 65.41 | 66.72 | **58.37** |

Table 3: Ablation study of the proposed components. **Calculate** denotes whether to compute the feature distance and include subset loss. **Compare** denotes whether to compare the feature distance. **Freeze** means whether to freeze preceding $\mathcal{S}_{\text{MLS}}$. CIFAR-10, IPC$_{10}$.

| | | 1 | 2 | 3 | 4 | 5 | 6 | 7 | 8 | 9 | 10 | Avg. | Diff. |
|---|---|---|---|---|---|---|---|---|---|---|---|---|---|
| ResNet | A | 41.6 | 49.6 | 52.5 | 54.8 | 57.9 | 59.0 | 61.0 | 61.4 | 62.1 | 63.7 | 56.4 | - |
| | B | 41.2 | 47.0 | 49.5 | 51.4 | 53.3 | 52.8 | 54.5 | 54.4 | 55.5 | 55.8 | 51.5 | - |
| | C | 25.5 | 34.0 | 39.1 | 45.9 | 53.0 | 54.6 | 57.3 | 59.7 | 61.7 | 63.7 | 49.5 | 0 |
| | Ours | 39.3 | 47.3 | 48.6 | 52.7 | 54.9 | 55.3 | 57.2 | 59.3 | 61.0 | 62.7 | 53.8 | +4.3 |
| DenseNet | A | 39.8 | 49.3 | 51.5 | 54.4 | 56.6 | 58.5 | 59.1 | 60.2 | 60.7 | 62.5 | 55.3 | - |
| | B | 41.0 | 48.1 | 48.3 | 52.3 | 54.8 | 53.0 | 55.3 | 53.5 | 54.1 | 55.3 | 51.6 | - |
| | C | 26.4 | 35.4 | 40.3 | 47.3 | 53.1 | 54.6 | 57.9 | 59.0 | 60.6 | 62.5 | 49.7 | 0 |
| | Ours | 38.5 | 46.8 | 49.6 | 53.4 | 54.6 | 56.3 | 57.5 | 58.4 | 59.7 | 61.0 | 53.6 | +3.9 |

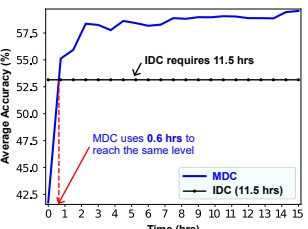

Table 4: Cross-architecture performance of the proposed method. CIFAR-10, IPC$_{10}$.

Figure 4: Accuracy vs. training time. CIFAR-10, IPC$_{10}$.

## 4.3 MORE ANALYSIS

**Ablation Study.** Tab. 3 provides the ablation study of the components for MLS selection. The **first row** is exactly the Baseline-C, which condenses once but does not include the subset loss. We can clearly observe the "subset degradation problem" in this case. **Row 2** includes the subset loss but does not consider "rate of change" for feature distance. In such a case, we find $\mathcal{S}_{\text{MLS}}=\mathcal{S}_{[1]}$ for all outer loops. We observe a large improvement for $\mathcal{S}_{[1]}$. However, the accuracy of $\mathcal{S}_{[2]}$ does not exceed $\mathcal{S}_{[1]}$ even though the dataset size is larger. **Row 3** does not consider the freezing strategy. It improves the average accuracy from 57.08% to 58.08% but makes the accuracy of $\mathcal{S}_{[1]}$ drop from 49.35% to 40.12%. **Row 4** is the proposed method's complete version, enjoying all components' benefits.

**Performance on Different Architectures.** Tab. 4 shows that the "subset degradation problem" is not unique to the ConvNet (Gidaris & Komodakis, 2018) model but also exists when using ResNet (He et al., 2016) and DenseNet (Huang et al., 2017). Not surprisingly, the proposed method generalizes to other models with improvement of +4.3% and +3.9% on the average accuracy for ResNet and DenseNet compared to Baseline-C, respectively.

**Performance on Different Condensation Method.** Apart from IDC, our MDC method can be applied to other basic condensation methods such as DREAM (Liu et al., 2023b). The average accuracy increases from 58.37% to 60.19%. Appendix B.3 shows the detailed accuracy numbers.

**Reduced Training Time Needed.** As shown in Fig. 4, the introduction of "Adaptive Subset Loss" increases our total training time from 11.5 hours (IDC) to 15.1 hours. However, our MDC method does not need such a long training time. As depicted by the red vertical line, our MDC method only needs **0.6 hours** to match IDC's average accuracy. In other words, we reduce the training time by 94.8% for the same performance. This might be because our "adaptive subset loss" provides extra supervision for the learning process. More details can be found in Appendix B.5.

**Subset Evaluation Metrics.** To evaluate the subsets, we have more metrics apart from the feature distance, which is the first component in Sec. 3.3.2. As shown in Tab. 5, we can also consider gradient distances or evaluate subsets based on model accuracy when trained on them. Our "Feature Distance" metric just needs two forward processes to calculate the feature of the subset $\mathcal{S}_{[N-1]}$ and real images. Compared to "Feature Distance", "Gradient Distance" demands more computational processes (ten forward and ten backward processes) but yields a similar accuracy. While using "Accuracy Difference" outperforms "Feature Distance", it is computationally intensive and impractical. Here, the required number of forward process **F** and backward process **B** is $E \times (N-1) = 1000 \times 9 = 9,000$.

**Visualization of MLS.** Selection of $\mathcal{S}_{\text{MLS}}$ is shown in Fig. 5. See Appendix B.6 for class-wise MLS.

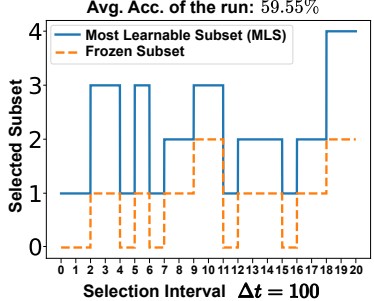

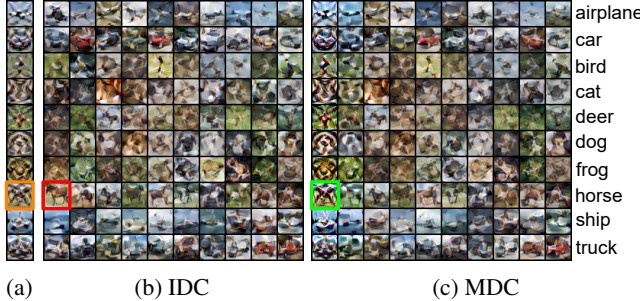

(a)    (b) IDC    (c) MDC

Figure 5: MLS and frozen subsets visualization. CIFAR-10, IPC$_{10}$.

Figure 6: (a) IDC condensed IPC$_1$; (b) IDC condensed IPC$_{10}$; (c) our MDC condensed IPC$_{10}$.

| Evaluation Metric | 1 | 2 | 3 | 4 | 5 | 6 | 7 | 8 | 9 | 10 | Avg. | **F** | **B** |
|---|---|---|---|---|---|---|---|---|---|---|---|---|---|
| Feature Distance | 49.66 | 54.58 | 53.92 | 54.55 | 55.18 | 58.80 | 61.51 | 63.36 | 65.41 | 66.72 | 58.37 | **2** | **0** |
| Gradient Distance | 49.20 | 53.64 | 56.48 | 56.37 | 55.82 | 59.53 | 61.05 | 63.31 | 65.00 | 66.90 | 58.73 | $N$ | $N$ |
| Accuracy Difference | 48.18 | 52.66 | 57.10 | 58.62 | 59.75 | 62.11 | 63.17 | 63.99 | 65.48 | 66.57 | 59.76 | $E\times(N-1)$ | $E\times(N-1)$ |

Table 5: Different subset evaluation metrics for MLS. **F** and **B** denote the number of forward and backward propagation, respectively. $E$ and $N$ denote the number of training epochs (i.e., 1000) and target IPC (i.e., 10), respectively. CIFAR-10, IPC$_{10}$.

| | Acc. | Diff. |
|---|---|---|
| Runs | 58.37 | +5.22 |
| | 58.49 | +5.34 |
| | 58.73 | +5.58 |
| | 58.90 | +5.75 |
| | 59.20 | +6.05 |
| | 59.55 | +6.40 |
| Avg. | 58.87 | +5.72 |

Table 6: Effects of condensation runs.

**Effects of Condensation Runs.** The variation in our results arises from two aspects: the condensation process and the model training using the condensed dataset. Tab. 6 shows that different condensation runs lead to improvements ranging from 5.22% to 6.40% over Baseline-C (53.15%). This variation is more substantial than the variation in model training (see Appendix B.4). Therefore, we report a lower result (58.37%) in Tab. 1 to illustrate the effectiveness of our method conservatively.

**Visualization of Condensed Dataset.** As shown in Fig. 6, we visualize and compare the dataset condensed with IDC (Kim et al., 2022b) and our MDC. We utilize three horse images from different settings to articulate our findings. The horse image in IDC condensed IPC$_1$ is highlighted with a yellow border, in the IDC condensed IPC$_{10}$ with a red border, and in our MDC condensed IPC$_1$ with a green border. For clarity, we'll refer to these as $horse_{orange}$, $horse_{red}$, and $horse_{green}$. **1)** Upon comparing $horse_{orange}$ with $horse_{red}$, it's evident that $horse_{orange}$ exhibits more distortion than $horse_{red}$ to save other images' information. **2)** Furthermore, when aligning $horse_{orange}$ with actual images (see Fig. 10 in Appendix C) , $horse_{orange}$ is almost the same as the real counterpart image, suggesting it doesn't include information from other images. This highlights the "subset degradation problem" – when a small subset, such as $horse_{red}$, lacks guidance during condensation, it fails to adequately represent the complete dataset. **3)** Upon evaluating $horse_{orange}$ against $horse_{green}$, we can observe that images condensed using our MDC approach display more pronounced distortion than those from IDC IPC$_1$. This increased distortion in $horse_{green}$ arises because it also serves as a subset for IPC$_2$, IPC$_3$, ... , IPC$_N$. This increased distortion demonstrates our method effectively addresses the "subset degradation problem". More visualization can be found in Appendix C.

## 5    CONCLUSION AND FUTURE WORK

To achieve multisize dataset condensation, our MDC method is the first to compress multiple condensation processes into a single condensation process. We adaptively select the most learnable subset (MLS) to build "adaptive subset loss" to mitigate the "subset degradation problem". Extensive experiments show that our method achieves state-of-the-art performance on the various models and datasets. Future works can include three directions. First, our subset loss has an impact on the accuracy of the full synthetic dataset, so we plan to find a way to maintain the accuracy better. Second, we aim to explore why our MDC learns much faster than previous methods. Third, better subset selection metrics are worth investigating.

## 6 ACKNOWLEDGEMENT

This work was supported in part by A*STAR Career Development Fund (CDF) under C233312004, in part by the National Research Foundation, Singapore, and the Maritime and Port Authority of Singapore / Singapore Maritime Institute under the Maritime Transformation Programme (Maritime AI Research Programme – Grant number SMI-2022-MTP-06).

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

## A  ALGORITHM

Algo. 1 provides the algorithm of the proposed MDC method.

---

**Algorithm 1** Multisize Dataset Condensation

---

**Input:** Full dataset $\mathcal{B}$, model $\Theta$, MLS selection period $\Delta t$, learning rate of the synthetic dataset $\lambda$, learning rate of the model $\eta$, outer loop iterations $T$, inner loop epochs $E$, and class loop iterations $C$.
**Output:** Synthetic dataset $\mathcal{S}$
 1: Initialize synthetic dataset $\mathcal{S}$
 2: Initialize the most learnable subset (MLS) $\mathcal{S}_{\text{MLS}}$
 3: **for** $t = 1$ **to** $T$ **do**                                                               ▷ Outer loop
 4:     Randomly initialize model weight $\theta_t$
 5:     **for** $e = 1$ **to** $E$ **do**                                                       ▷ Inner loop
 6:         **for** $c = 1$ **to** $C$ **do**                                                   ▷ Class loop
 7:             Sample class-wise mini-batches $B_c \sim \mathcal{B}$, $S_c \sim \mathcal{S}$
 8:             Update $\mathcal{S}_c$ with **subset loss** according to Eq. 10
 9:         **end for**
10:         $\theta_{t,e+1} \leftarrow \theta_{t,e} - \eta \nabla_\theta \ell\left(\theta_{t,e}; B\right)$      ▷ Update model with real image mini-batch $B \sim \mathcal{B}$
11:     **end for**
12:     **if** $t \% \Delta t$ is 0 **then**                                                ▷ Every $\Delta t$ iterations
13:         Select $\mathcal{S}_{\text{MLS}}$ according to Eq. 9
14:     **end if**
15: **end for**
16: **return** Synthetic dataset $\mathcal{S}$

---

## B  EXPERIMENT

### B.1  EXPERIMENT SETTINGS

**Datasets:**

- SVHN (Netzer et al., 2011) contains street digits of shape $32 \times 32 \times 3$. The dataset contains 10 classes including digits from 0 to 9. The training set has 73257 images, and the test set has 26032 images.

- CIFAR-10 (Krizhevsky et al., 2009) contains images of shape $32 \times 32 \times 3$ and has 10 classes in total: airplane, automobile, bird, cat, deer, dog, frog, horse, ship, and truck. The training set has 5,000 images per class and the test set has 1,000 images per class, containing in total 50,000 training images and 10,000 testing images.

- CIFAR-100 (Krizhevsky et al., 2009) contains images of shape $32 \times 32 \times 3$ and has 100 classes in total. Each class contains 500 images for training and 100 images for testing, leading to a total of 50,000 training images and 10,000 testing images.

- ImageNet-10 (Deng et al., 2009) is a subset of ImageNet-1K (Deng et al., 2009) containing images with an average $469 \times 387 \times 3$ pixels but reshaped to resolution of $224 \times 224 \times 3$. It contains 1,280 training images per class on average and a total of 50,000 images for testing (validation set). Following Kim et al. (2022b), the ImageNet-10 contains 10 classes: 1) poke bonnet, 2) green mamba, 3) langur, 4) Doberman pinscher, 5) gyromitra, 6) gazelle hound, 7) vacuum cleaner, 8) window screen, 9) cocktail shaker, and 10) garden spider.

**Augmentation:** Following IDC (Kim et al., 2022b), we perform augmentation during training networks in condensation and evaluation, and we use coloring, cropping, flipping, scaling, rotating, and mixup. When updating network parameters, image augmentations are different for each image in a batch; when updating synthetic images, the same augmentations are utilized for the synthetic images and corresponding real images in a batch.

- Color which adjusts the brightness, saturation, and contrast of images.

- Crop which pads the image and then randomly crops back to the original size.
- Flip which flips the images horizontally with a probability of 0.5.
- Scale which randomly scales the images by a factor according to a ratio.
- Rotate which rotates the image by a random angle according to a ratio.
- Cutout which randomly removes square parts of the image, replacing the removed parts with black squares.
- Mixup which randomly selects a square region within the image and replaces this region with the corresponding section from another randomly chosen image. It happens at a probability of 0.5.

**Multi-formation Settings.** For all results we use IDC (Kim et al., 2022b) as the "basic condensation method" otherwise stated. Following its setup, we use a multi-formation factor of 2 for SVHN, CIFAR-10, CIFAR-100 datasets and a factor of 3 for ImageNet-10.

**Reason for Using Large Batch Size for IPC > 32.** For CIFAR-10, CIFAR-100 and SVHN, we use the default batch size (128) when IPC $< 32$ and a larger batch size (256) when $32 \leq$ IPC $\leq 64$. The reason is that our method is based on IDC (Kim et al., 2022b) which uses a multi-formation factor of $f = 2$ for CIFAR-10, CIFAR-100, and SVHN datasets. The multi-formation function splits a synthetic image into $f^2 = 2^2 = 4$ images during the condensation process. To ensure all samples in a subset can be sampled during condensation, we increase the subsets when the number of images exceeds the default batch size, which is 128. With a multi-formation factor $f = 2$, the maximum IPC of each sampling process is IPC $= 32$ (i.e., IPC $\times 2^2 \leq 128$). For ImageNet-10 IPC$_{20}$, a multi-formation factor of $f = 3$ is used. Hence, we use a batch size of 256 (i.e., $128 \leq 20 \times 3^2 \leq 256$).

**MTT Settings.** The reported numbers of MTT (Cazenavette et al., 2022) are obtained without ZCA normalization to keep all methods using the standard normalization technique.

## B.2 FEATURE DISTANCE CALCULATION

Tab. 7 presents the feature distance computed at a specific outer loop $t$ without imposing the subset loss. The table conveys two pieces of information. First, the feature loss of a smaller subset is always greater than that of a larger subset. That is a reason why we need to find the rate of change. Otherwise, $S_{[1]}$ will always be selected. Second, the feature distance of the smallest subset changes the most, and this contributes to why we select $S_{[1]}$ as the subset initialization.

|  | 1 | 2 | 3 | 4 | 5 | 6 | 7 | 8 | 9 |
|---|---|---|---|---|---|---|---|---|---|
| $t = 1$ | 3012 | 1678 | 1249 | 1013 | 896 | 807 | 738 | 701 | 675 |
| $t = 50$ | 2596 | 1294 | 891 | 661 | 514 | 429 | 373 | 332 | 298 |
| Diff. | 416 | 384 | 358 | 352 | 382 | 378 | 365 | 369 | 377 |
| Rate of change | 8.32 | 7.68 | 7.16 | 7.04 | 7.64 | 7.56 | 7.30 | 7.38 | 7.54 |

Table 7: Feature distance of subsets summed over inner loop training. CIFAR-10, IPC$_{10}$.

## B.3 THE INFLUENCE OF BASIC CONDENSATION METHOD

Tab. 8 shows our method works on other basic condensation methods such as DREAM (Liu et al., 2023b).

|  | 1 | 2 | 3 | 4 | 5 | 6 | 7 | 8 | 9 | 10 | Avg. |
|---|---|---|---|---|---|---|---|---|---|---|---|
| IDC | 49.66 | 54.58 | 53.92 | 54.55 | 55.18 | 58.80 | 61.51 | 63.36 | 65.41 | 66.72 | 58.37 |
| DREAM | 49.70 | 55.12 | 55.84 | 57.59 | 58.72 | 62.92 | 63.61 | 64.71 | 66.26 | 67.44 | 60.19 |

Table 8: Comparison between the proposed method applied to IDC Kim et al. (2022b) and to DREAM Liu et al. (2023b) on CIFAR-10 IPC$_{10}$.

## B.4 PRIMARY RESULTS WITH STANDARD DEVIATION

In Tab. 9, we list the primary results with standard deviation for synthetic datasets with $IPC_{10}$ and $IPC_{50}$, including SVHN, CIFAR-10, and CIFAR-100 datasets. The standard deviation is computed from three randomly initialized networks since the same subset is selected for each run. Even by taking into account these standard deviations, our method shows a consistent improvement in the average accuracy.

| Dataset | | 1 | 2 | 3 | 4 | 5 | 6 | 7 | 8 | 9 | 10 | Avg. | Diff. |
|---|---|---|---|---|---|---|---|---|---|---|---|---|---|
| SVHN | A | $68.50^{\dagger}_{\pm0.9}$ | $75.27_{\pm0.3}$ | $79.55_{\pm0.4}$ | $81.85_{\pm0.2}$ | $83.33_{\pm0.1}$ | $84.53_{\pm0.3}$ | $85.66_{\pm0.3}$ | $86.05_{\pm0.1}$ | $86.25_{\pm0.2}$ | $87.50^{\dagger}_{\pm0.3}$ | 81.85 | - |
| | B | $68.50^{\dagger}_{\pm0.9}$ | $71.65_{\pm0.1}$ | $71.27_{\pm0.9}$ | $71.92_{\pm0.3}$ | $73.28_{\pm0.3}$ | $70.74_{\pm0.4}$ | $71.83_{\pm0.4}$ | $71.08_{\pm0.8}$ | $71.97_{\pm1.0}$ | $71.55_{\pm0.7}$ | 71.38 | - |
| | C | $35.48_{\pm0.4}$ | $51.55_{\pm0.6}$ | $60.42_{\pm1.0}$ | $67.97_{\pm0.5}$ | $74.38_{\pm0.5}$ | $77.65_{\pm0.7}$ | $81.70_{\pm0.2}$ | $83.86_{\pm0.5}$ | $85.96_{\pm0.4}$ | $87.50^{\dagger}_{\pm0.3}$ | 70.65 | 0 |
| | Ours | $63.26_{\pm1.0}$ | $67.91_{\pm0.7}$ | $72.15_{\pm1.0}$ | $74.09_{\pm0.3}$ | $77.54_{\pm0.4}$ | $78.17_{\pm0.3}$ | $80.92_{\pm0.3}$ | $82.82_{\pm0.5}$ | $84.27_{\pm0.3}$ | $86.38_{\pm0.2}$ | 76.75 | +6.10 |
| CIFAR-10 | A | $50.80_{\pm0.3}$ | $54.85_{\pm0.4}$ | $59.79_{\pm0.2}$ | $61.84_{\pm0.1}$ | $62.49_{\pm0.3}$ | $64.59_{\pm0.1}$ | $65.53_{\pm0.2}$ | $66.33_{\pm0.1}$ | $66.82_{\pm0.3}$ | $67.50^{\dagger}_{\pm0.5}$ | 62.05 | - |
| | B | $50.80_{\pm0.3}$ | $53.17_{\pm0.4}$ | $55.09_{\pm0.4}$ | $56.17_{\pm0.3}$ | $55.80_{\pm0.3}$ | $56.98_{\pm0.3}$ | $57.60_{\pm0.3}$ | $57.78_{\pm0.1}$ | $58.22_{\pm0.4}$ | $58.38_{\pm0.1}$ | 56.00 | - |
| | C | $27.49_{\pm0.8}$ | $38.50_{\pm0.5}$ | $45.29_{\pm0.1}$ | $50.85_{\pm0.4}$ | $53.60_{\pm0.3}$ | $57.98_{\pm0.2}$ | $60.99_{\pm0.5}$ | $63.60_{\pm0.2}$ | $65.71_{\pm0.1}$ | $67.50^{\dagger}_{\pm0.5}$ | 53.15 | 0 |
| | Ours | $49.66_{\pm0.4}$ | $54.58_{\pm0.2}$ | $53.92_{\pm0.3}$ | $54.55_{\pm0.2}$ | $55.18_{\pm0.2}$ | $58.80_{\pm0.5}$ | $61.51_{\pm0.3}$ | $63.36_{\pm0.2}$ | $65.41_{\pm0.3}$ | $66.72_{\pm0.1}$ | 58.37 | +5.22 |
| CIFAR-100 | A | $28.90^{\dagger}_{\pm0.2}$ | $34.28_{\pm0.2}$ | $37.35_{\pm0.2}$ | $39.13_{\pm0.1}$ | $41.15_{\pm0.4}$ | $42.65_{\pm0.4}$ | $43.62_{\pm0.3}$ | $44.48_{\pm0.2}$ | $45.07_{\pm0.1}$ | $45.40_{\pm0.4}$ | 40.20 | - |
| | B | $28.90^{\dagger}_{\pm0.2}$ | $30.63_{\pm0.1}$ | $31.64_{\pm0.0}$ | $31.76_{\pm0.2}$ | $32.61_{\pm0.2}$ | $32.85_{\pm0.2}$ | $33.03_{\pm0.3}$ | $33.04_{\pm0.2}$ | $33.32_{\pm0.2}$ | $33.39_{\pm0.2}$ | 32.12 | - |
| | C | $14.38_{\pm0.2}$ | $21.76_{\pm0.2}$ | $28.01_{\pm0.2}$ | $32.21_{\pm0.3}$ | $35.27_{\pm0.3}$ | $39.09_{\pm0.2}$ | $40.92_{\pm0.1}$ | $42.69_{\pm0.2}$ | $44.28_{\pm0.2}$ | $45.40_{\pm0.4}$ | 34.40 | 0 |
| | Ours | $27.58_{\pm0.2}$ | $31.83_{\pm0.0}$ | $33.59_{\pm0.2}$ | $35.42_{\pm0.1}$ | $36.93_{\pm0.1}$ | $38.95_{\pm0.4}$ | $40.70_{\pm0.1}$ | $42.05_{\pm0.1}$ | $43.86_{\pm0.1}$ | $44.34_{\pm0.2}$ | 37.53 | +3.13 |

(a) Results of SVHN, CIFAR-10, CIFAR-100 targeting $IPC_{10}$.

| Dataset | | 1 | 2 | 3 | 4 | 5 | 6 | 7 | 8 | 9 | 10 |
|---|---|---|---|---|---|---|---|---|---|---|---|
| SVHN | A | $68.50^{\dagger}_{\pm0.9}$ | $75.27_{\pm0.3}$ | $79.55_{\pm0.4}$ | $81.85_{\pm0.2}$ | $83.33_{\pm0.1}$ | $84.53_{\pm0.3}$ | $85.66_{\pm0.3}$ | $86.05_{\pm0.1}$ | $86.25_{\pm0.2}$ | $87.50^{\dagger}_{\pm0.3}$ |
| | C | $34.90_{\pm0.9}$ | $46.52_{\pm0.4}$ | $52.23_{\pm0.9}$ | $56.30_{\pm0.4}$ | $62.25_{\pm0.5}$ | $65.34_{\pm0.5}$ | $68.84_{\pm0.3}$ | $69.57_{\pm1.7}$ | $71.95_{\pm0.5}$ | $74.69_{\pm0.2}$ |
| | Ours | $58.77_{\pm1.5}$ | $67.72_{\pm0.3}$ | $69.33_{\pm0.5}$ | $72.26_{\pm0.4}$ | $75.02_{\pm0.3}$ | $73.71_{\pm0.7}$ | $74.50_{\pm0.5}$ | $74.63_{\pm0.6}$ | $76.21_{\pm0.4}$ | $76.87_{\pm0.7}$ |
| CIFAR-10 | A | $50.80_{\pm0.3}$ | $54.85_{\pm0.4}$ | $59.79_{\pm0.2}$ | $61.84_{\pm0.1}$ | $62.49_{\pm0.3}$ | $64.59_{\pm0.1}$ | $65.53_{\pm0.2}$ | $66.33_{\pm0.1}$ | $66.82_{\pm0.3}$ | $67.50^{\dagger}_{\pm0.5}$ |
| | C | $27.87_{\pm0.4}$ | $35.69_{\pm0.5}$ | $41.93_{\pm0.2}$ | $45.29_{\pm0.2}$ | $47.54_{\pm0.4}$ | $51.96_{\pm0.4}$ | $53.51_{\pm0.3}$ | $55.59_{\pm0.1}$ | $56.62_{\pm0.2}$ | $58.26_{\pm0.1}$ |
| | Ours | $47.83_{\pm0.6}$ | $52.18_{\pm0.2}$ | $56.29_{\pm0.1}$ | $58.52_{\pm0.2}$ | $58.75_{\pm0.4}$ | $60.67_{\pm0.3}$ | $61.90_{\pm0.1}$ | $62.74_{\pm0.2}$ | $62.32_{\pm0.2}$ | $62.64_{\pm0.2}$ |
| CIFAR-100 | A | $28.90^{\dagger}_{\pm0.2}$ | $34.28_{\pm0.2}$ | $37.35_{\pm0.2}$ | $39.13_{\pm0.1}$ | $41.15_{\pm0.4}$ | $42.65_{\pm0.4}$ | $43.62_{\pm0.3}$ | $44.48_{\pm0.2}$ | $45.07_{\pm0.1}$ | $45.40_{\pm0.4}$ |
| | C | $12.66_{\pm0.1}$ | $18.35_{\pm0.1}$ | $23.76_{\pm0.4}$ | $26.92_{\pm0.4}$ | $29.12_{\pm0.2}$ | $32.23_{\pm0.1}$ | $34.21_{\pm0.4}$ | $35.71_{\pm0.3}$ | $37.18_{\pm0.3}$ | $38.25_{\pm0.3}$ |
| | Ours | $26.34_{\pm0.2}$ | $29.71_{\pm0.3}$ | $31.74_{\pm0.4}$ | $32.95_{\pm0.4}$ | $34.49_{\pm0.3}$ | $36.36_{\pm0.2}$ | $38.49_{\pm0.4}$ | $39.59_{\pm0.1}$ | $40.43_{\pm0.4}$ | $41.35_{\pm0.2}$ |

| Dataset | | 20 | 30 | 40 | 50 | Avg. | Diff. |
|---|---|---|---|---|---|---|---|
| SVHN | A | $89.54_{\pm0.2}$ | $90.27_{\pm0.1}$ | $91.09_{\pm0.1}$ | $91.38_{\pm0.1}$ | 84.34 | - |
| | C | $83.73_{\pm0.1}$ | $87.83_{\pm0.1}$ | $89.73_{\pm0.0}$ | $91.38_{\pm0.1}$ | 68.23 | 0 |
| | Ours | $83.67_{\pm0.2}$ | $87.08_{\pm0.2}$ | $89.46_{\pm0.2}$ | $91.39_{\pm0.1}$ | 76.47 | +8.24 |
| CIFAR-10 | A | $70.82_{\pm0.3}$ | $72.86_{\pm0.5}$ | $74.30_{\pm0.0}$ | $75.07_{\pm0.2}$ | 65.26 | - |
| | C | $66.77_{\pm0.1}$ | $70.50_{\pm0.2}$ | $72.98_{\pm0.3}$ | $74.50_{\pm0.2}$ | 54.21 | 0 |
| | Ours | $66.88_{\pm0.2}$ | $70.02_{\pm0.2}$ | $72.91_{\pm0.5}$ | $74.56_{\pm0.3}$ | 62.01 | +7.80 |
| CIFAR-100 | A | $49.50_{\pm0.5}$ | $52.28_{\pm0.3}$ | $52.54_{\pm0.3}$ | $53.47_{\pm0.5}$ | 43.56 | - |
| | C | $45.67_{\pm0.3}$ | $49.60_{\pm0.2}$ | $52.36_{\pm0.1}$ | $53.47_{\pm0.5}$ | 34.96 | 0 |
| | Ours | $46.06_{\pm0.3}$ | $49.40_{\pm0.1}$ | $51.72_{\pm0.1}$ | $53.67_{\pm0.4}$ | 39.45 | +4.49 |

(b) Results of SVHN, CIFAR-10, CIFAR-100 targeting $IPC_{50}$.

Table 9: Comparisons between the proposed method and three different baselines built with the IDC Kim et al. (2022b). $^{\dagger}$ represents the numbers reported in the original paper. Results from sub-table (b) are divided into two parts due to limited space.

### B.5 ACCURACY OF SUBSETS DURING CONDENSATION PROCESS

Using different condensation runs in Tab. 6, we analyze the effect of subset selection during condensation. As shown in Fig. 7, the run with an accuracy of 58.37% has only IPC-{1,2} are selected as $\mathcal{S}_{\text{MLS}}$. Tab. 10 shows the accuracies for each subset during the condensation process. As shown in Fig. 8, the run with an accuracy of 59.55% has IPC-{1,2,3,4} are selected as $\mathcal{S}_{\text{MLS}}$. Tab. 11 shows the accuracies for each subset during the condensation process. We hypothesize that optimizing a subset with a larger IPC during the condensation process will lead to higher accuracy as it provides more supervision signals to guide the subset optimization.

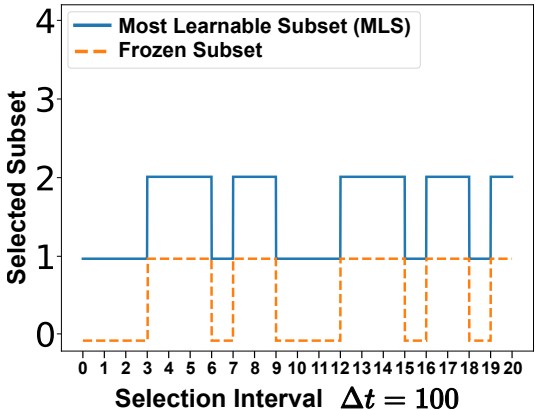

Figure 7: Visualization for the run with $\mathcal{S}_{\text{MLS}}$ including $\text{IPC}_1$ and $\text{IPC}_2$.

|      | 100   | 200   | 300   | 400   | 500   | 600   | 700   | 800   | 900   | 1000  |
|------|-------|-------|-------|-------|-------|-------|-------|-------|-------|-------|
| 1    | 48.79 | 48.62 | 48.16 | 48.81 | 49.58 | 49.03 | 49.22 | 49.37 | 49.19 | 49.75 |
| 2    | 47.24 | 46.45 | 46.56 | 47.02 | 53.94 | 53.76 | 53.93 | 54.20 | 54.49 | 53.93 |
| 3    | 47.95 | 48.14 | 48.04 | 47.93 | 54.02 | 53.73 | 53.91 | 53.74 | 53.81 | 53.82 |
| 4    | 51.07 | 50.54 | 50.98 | 50.82 | 54.31 | 54.17 | 53.98 | 54.16 | 54.55 | 54.51 |
| 5    | 52.92 | 52.05 | 52.18 | 52.08 | 54.84 | 54.53 | 54.77 | 54.38 | 55.17 | 55.26 |
| 6    | 56.03 | 55.58 | 56.01 | 55.74 | 57.63 | 57.99 | 57.78 | 58.14 | 58.73 | 58.10 |
| 7    | 58.09 | 58.27 | 57.92 | 57.81 | 60.46 | 60.56 | 60.97 | 60.14 | 60.97 | 60.90 |
| 8    | 59.72 | 60.31 | 60.03 | 59.83 | 62.43 | 62.21 | 62.17 | 62.27 | 62.82 | 62.68 |
| 9    | 61.88 | 61.90 | 61.60 | 62.13 | 63.79 | 63.95 | 64.31 | 63.97 | 64.64 | 64.54 |
| 10   | 63.21 | 63.19 | 63.52 | 63.00 | 65.79 | 65.59 | 65.70 | 65.38 | 66.35 | 66.28 |
| Avg. | 54.69 | 54.51 | 54.50 | 54.52 | 57.68 | 57.55 | 57.67 | 57.57 | 58.07 | 57.98 |
|      | 1100  | 1200  | 1300  | 1400  | 1500  | 1600  | 1700  | 1800  | 1900  | 2000  |
| 1    | 49.75 | 49.76 | 49.25 | 49.18 | 49.76 | 49.64 | 49.61 | 49.63 | 49.44 | 49.66 |
| 2    | 54.38 | 53.87 | 54.47 | 53.33 | 53.77 | 53.87 | 54.72 | 54.82 | 54.55 | 54.58 |
| 3    | 53.52 | 53.46 | 53.51 | 53.39 | 53.94 | 53.56 | 54.69 | 54.00 | 54.27 | 53.92 |
| 4    | 54.67 | 54.80 | 54.72 | 54.71 | 54.34 | 54.51 | 54.34 | 54.66 | 54.73 | 54.55 |
| 5    | 54.94 | 55.04 | 54.58 | 54.60 | 55.04 | 54.63 | 55.25 | 54.57 | 55.20 | 55.18 |
| 6    | 58.40 | 58.68 | 58.16 | 58.06 | 58.83 | 58.45 | 58.97 | 58.65 | 58.92 | 58.80 |
| 7    | 60.83 | 60.77 | 61.08 | 61.14 | 61.06 | 61.13 | 60.99 | 61.45 | 61.07 | 61.51 |
| 8    | 62.78 | 62.76 | 62.68 | 62.95 | 63.04 | 62.88 | 63.23 | 63.65 | 63.90 | 63.36 |
| 9    | 64.91 | 64.91 | 64.99 | 64.52 | 64.87 | 64.86 | 64.98 | 65.29 | 65.05 | 65.41 |
| 10   | 66.04 | 65.90 | 66.37 | 66.69 | 66.56 | 66.41 | 66.76 | 67.10 | 66.40 | 66.72 |
| Avg. | 58.02 | 57.99 | 57.98 | 57.86 | 58.12 | 57.99 | 58.35 | 58.38 | 58.35 | 58.37 |

Table 10: Accuracy of subsets evaluated at different outer loops. Selected $\mathcal{S}_{\text{MLS}}$ includes $\text{IPC}_1$ and $\text{IPC}_2$. CIFAR-10, $\text{IPC}_{10}$.

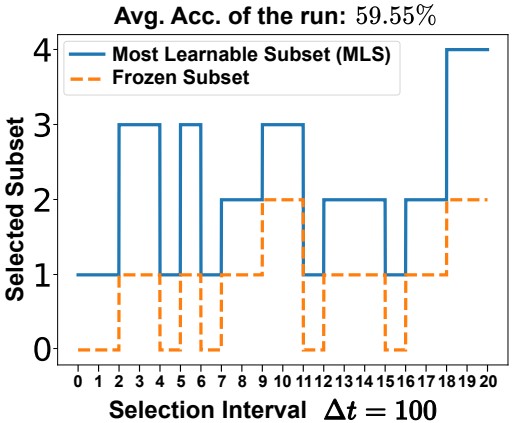

Figure 8: Visualization for the run with $\mathcal{S}_{\text{MLS}}$ including IPC$_1$, IPC$_2$, IPC$_3$, and IPC$_4$.

|      | 100   | 200   | 300   | 400   | 500   | 600   | 700   | 800   | 900   | 1000  |
|------|-------|-------|-------|-------|-------|-------|-------|-------|-------|-------|
| 1    | 49.40 | 50.20 | 50.10 | 51.00 | 50.00 | 50.60 | 49.80 | 48.70 | 48.90 | 49.74 |
| 2    | 47.30 | 47.60 | 50.30 | 50.10 | 48.20 | 49.50 | 49.50 | 53.10 | 53.60 | 53.41 |
| 3    | 48.40 | 48.60 | 57.80 | 57.90 | 55.30 | 57.20 | 55.80 | 55.70 | 56.00 | 57.46 |
| 4    | 52.30 | 51.80 | 57.40 | 57.60 | 55.00 | 57.60 | 56.60 | 55.60 | 55.90 | 56.93 |
| 5    | 52.80 | 53.60 | 56.30 | 56.50 | 56.20 | 57.10 | 56.30 | 55.80 | 56.20 | 56.59 |
| 6    | 57.00 | 57.90 | 59.60 | 58.50 | 59.20 | 60.10 | 60.00 | 59.00 | 58.50 | 59.58 |
| 7    | 58.10 | 59.50 | 61.10 | 60.00 | 61.50 | 61.50 | 61.30 | 60.70 | 60.50 | 61.24 |
| 8    | 60.50 | 61.90 | 62.30 | 62.20 | 62.50 | 63.30 | 63.10 | 63.10 | 62.50 | 63.13 |
| 9    | 62.00 | 63.10 | 64.10 | 64.20 | 63.90 | 64.40 | 65.30 | 64.20 | 64.30 | 64.72 |
| 10   | 63.40 | 65.10 | 64.60 | 64.50 | 65.90 | 64.90 | 66.50 | 65.80 | 66.40 | 66.03 |
| Avg. | 55.12 | 55.93 | 58.36 | 58.25 | 57.77 | 58.62 | 58.42 | 58.17 | 58.28 | 58.88 |
|      | 1100  | 1200  | 1300  | 1400  | 1500  | 1600  | 1700  | 1800  | 1900  | 2000  |
| 1    | 49.19 | 49.58 | 49.36 | 49.38 | 49.41 | 49.68 | 49.68 | 49.14 | 49.41 | 49.55 |
| 2    | 53.40 | 53.05 | 54.14 | 53.78 | 53.93 | 53.11 | 53.48 | 53.79 | 53.86 | 53.75 |
| 3    | 57.00 | 57.19 | 57.03 | 57.51 | 56.61 | 56.74 | 56.58 | 56.54 | 56.20 | 56.39 |
| 4    | 57.12 | 57.38 | 56.81 | 57.17 | 57.17 | 56.75 | 56.88 | 56.51 | 58.88 | 59.33 |
| 5    | 56.88 | 56.49 | 56.84 | 56.80 | 56.72 | 56.33 | 56.33 | 56.64 | 58.05 | 58.13 |
| 6    | 59.58 | 59.37 | 59.29 | 59.62 | 59.62 | 59.38 | 59.59 | 59.22 | 60.55 | 60.62 |
| 7    | 61.35 | 61.51 | 61.41 | 61.20 | 61.66 | 61.48 | 61.32 | 61.13 | 61.72 | 62.06 |
| 8    | 62.78 | 63.93 | 63.38 | 63.55 | 63.20 | 63.64 | 63.41 | 63.18 | 63.41 | 63.59 |
| 9    | 64.75 | 64.68 | 65.12 | 65.36 | 65.14 | 65.32 | 64.88 | 65.52 | 65.40 | 65.25 |
| 10   | 66.15 | 66.56 | 66.22 | 66.22 | 66.80 | 66.36 | 66.64 | 66.88 | 66.91 | 66.79 |
| Avg. | 58.82 | 58.97 | 58.96 | 59.06 | 59.03 | 58.88 | 58.88 | 58.86 | 59.44 | 59.55 |

Table 11: Accuracy of subsets evaluated at different outer loops. Selected $\mathcal{S}_{\text{MLS}}$ includes IPC$_1$, IPC$_2$, IPC$_3$, and IPC$_4$. CIFAR-10, IPC$_{10}$.

### B.6    CLASS-WISE MLS SELECTION

**Stable to Class-wise and Non-class-wise.** By default, we employ a uniform MLS size across all image classes for simplicity. However, our approach can be easily extended to maintain class-specific MLS sizes. As indicated in Tab. 12, our approach performs consistently in both class-wise and non-class-wise settings.

**Visualization of Class-wise MLS Selection.** Fig. 6 presents the choice of MLS of each class at every selection round. Compared to the non-class-wise manner (Fig. 3b), the class-wise manner selection tends to select relatively larger subsets.

|  | class-wise | 1 | 2 | 3 | 4 | 5 | 6 | 7 | 8 | 9 | 10 | 20 | 30 | 40 | 50 | Avg. |
|---|---|---|---|---|---|---|---|---|---|---|---|---|---|---|---|---|
| $IPC_{10}$ | ✓ | 49.22 | 52.90 | 56.13 | 56.98 | 57.55 | 61.05 | 62.22 | 63.57 | 65.44 | 66.90 | - | - | - | - | **59.19** |
|  | - | 49.66 | 54.58 | 53.92 | 54.55 | 55.18 | 58.80 | 61.51 | 63.36 | 65.41 | 66.72 | - | - | - | - | 58.37 |
| $IPC_{50}$ | ✓ | 48.17 | 53.35 | 55.68 | 57.11 | 56.75 | 59.57 | 60.02 | 60.31 | 60.76 | 61.55 | 66.79 | 70.29 | 72.77 | 74.57 | 61.26 |
|  | - | 47.83 | 52.18 | 56.29 | 58.52 | 58.75 | 60.67 | 61.90 | 62.74 | 62.32 | 62.64 | 66.88 | 70.02 | 72.91 | 74.56 | **62.01** |

Table 12: Class-wise v.s. non-class-wise $\mathcal{S}_{\text{MLS}}$ . CIFAR-10.

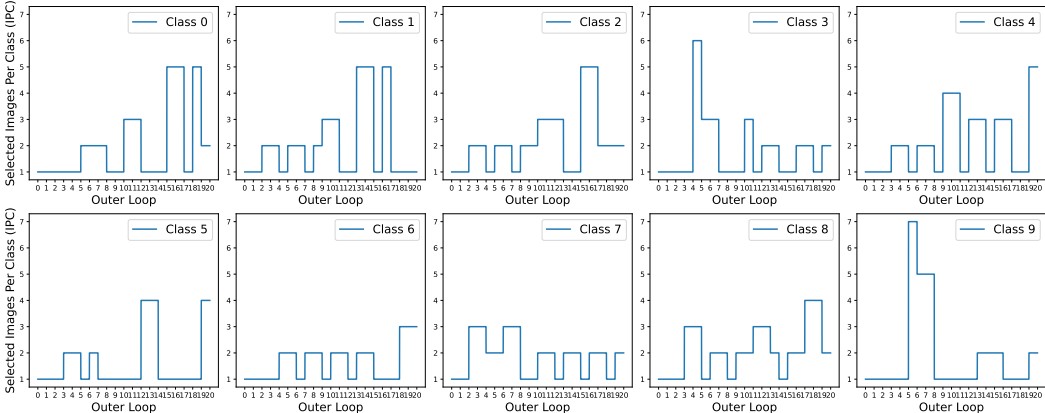

Figure 9: Visualization of selected subsets using a class-wise approach.

## C   VISUALIZATION OF CONDENSED IMAGES

### C.1   CIFAR-10

Fig. 10, 11 show the effectiveness of the proposed method. Note that the two figures using a multi-formation factor of 1 are for the purpose of better visualization. All experimental results shown in this visualization use the same settings as the main results reported in Tab. 1. Fig. 12 presents the visualizations of MDC on the CIFAR-10 dataset using a factor of 2 (Kim et al., 2022b).

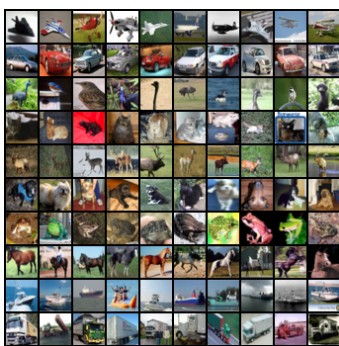

Figure 10: Visualization of the initialization of CIFAR-10, $IPC_{10}$.

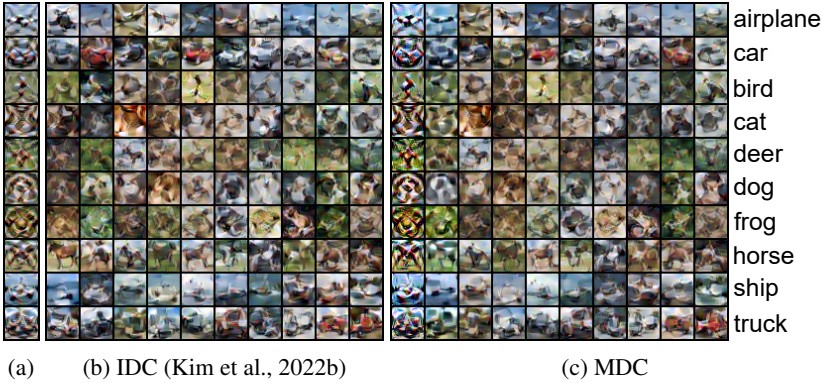

    (a)        (b) IDC (Kim et al., 2022b)         (c) MDC

Figure 11: Visualization of the proposed condensation method. (a) and (b) are IDC (Kim et al., 2022b) condensed to $IPC_1$ and $IPC_{10}$, respectively. (c) is the proposed method, MDC. CIFAR-10.

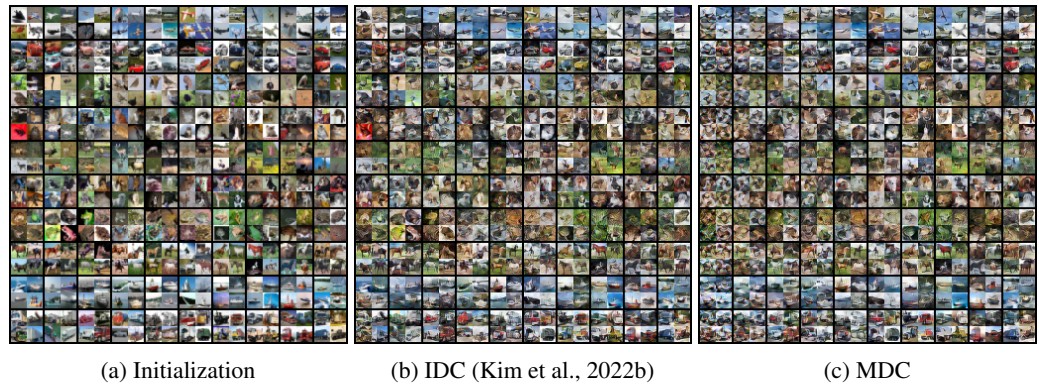

    (a) Initialization        (b) IDC (Kim et al., 2022b)        (c) MDC

Figure 12: Visualization of the proposed method. The number of the multi-formation factor is 2, meaning each condensed image is a composite of four original images. CIFAR-10, $IPC_{10}$.

## C.2 CIFAR-100

Fig. 13 visualizes the effects of MDC on the CIFAR-100 dataset using a factor of 2 (Kim et al., 2022b).

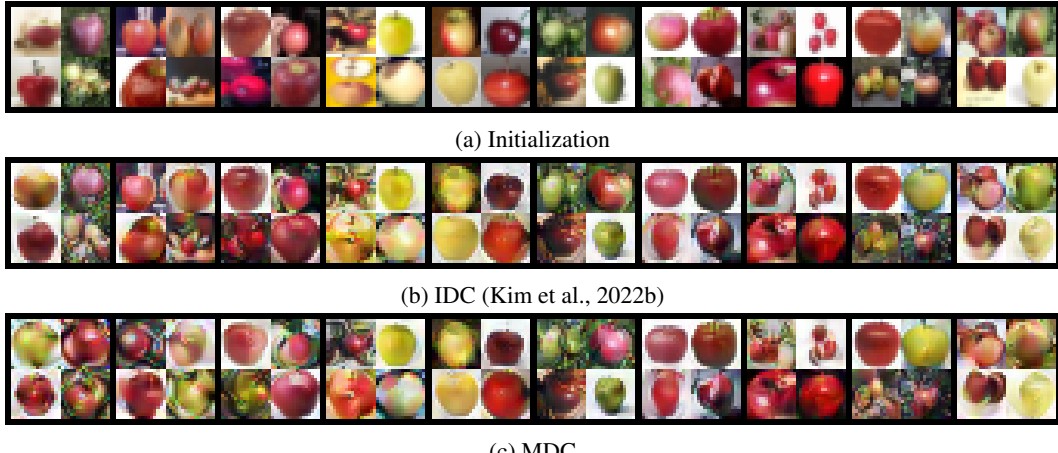

Figure 13: Visualization of the proposed condensation method. The number of the multi-formation factor is 2, meaning each condensed image is a composite of four original images. CIFAR-100, $IPC_{10}$, class: apple.

## C.3 SVHN

Fig. 14 presents the visualizations of MDC on the SVHN dataset using a factor of 2 (Kim et al., 2022b).

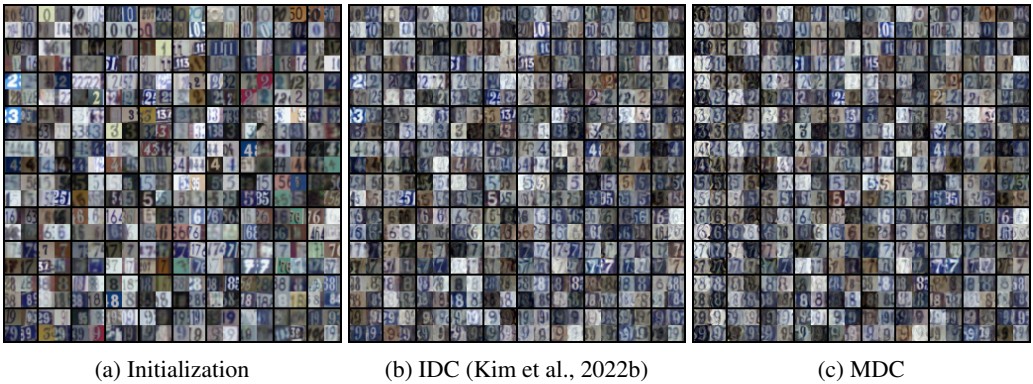

Figure 14: Visualization of the proposed condensation method. The number of the multi-formation factor is 2, meaning each condensed image is a composite of four original images. SVHN, $IPC_{10}$.

## C.4 IMAGENET

Fig. 15 uses a factor of 3 for ImageNet. Through comparing the images (class: gazelle hound) highlighted by orange, red and green boxes in Fig. 15, we observe the similar pattern shown in Fig. 6 that our MDC has large distortion.

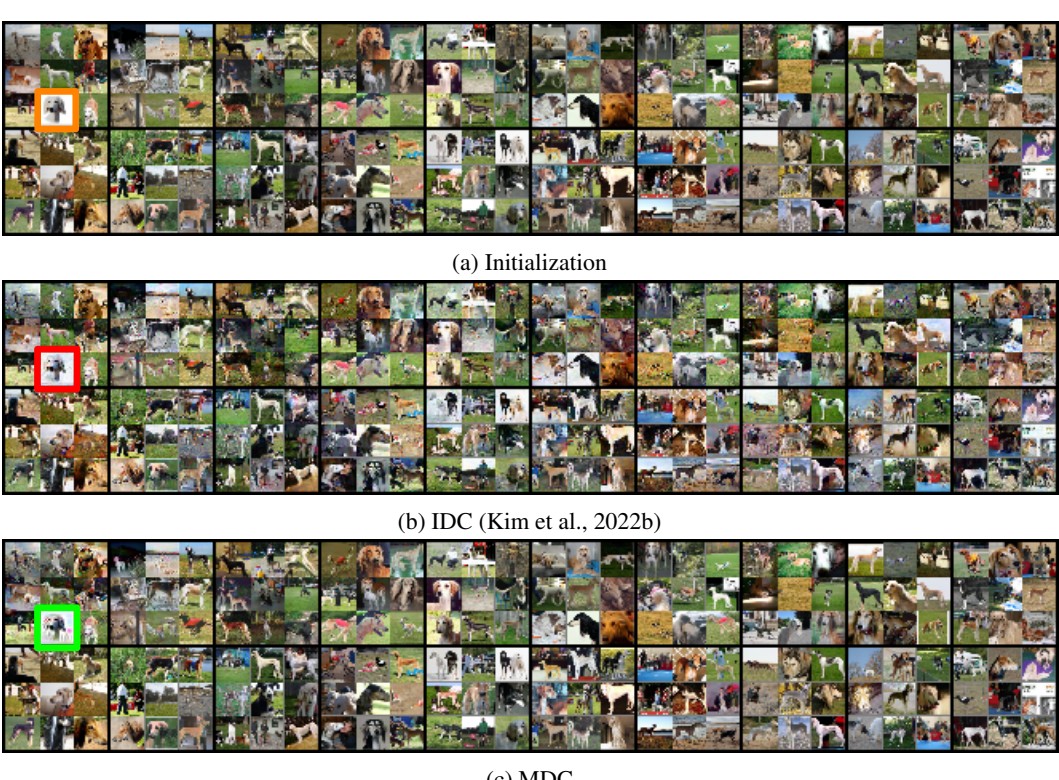

(a) Initialization

(b) IDC (Kim et al., 2022b)

(c) MDC

Figure 15: Visualization of on ImageNet targeting $IPC_{20}$. The number of the multi-formation factor is 3, meaning each condensed image is a composite of nine original images. class: gazelle hound.

