# OpenReview forum: "Multisize Dataset Condensation"
_ICLR.cc/2024/Conference — ICLR 2024 oral_

### Official Review · Reviewer_DoMB · 2023-10-27

**Soundness:** 4 excellent
**Presentation:** 4 excellent
**Contribution:** 4 excellent
**Rating:** 8
**Confidence:** 5

**Summary:**

This paper proposes a novel method to compress the condensation process into one process. It is different from the model compression or dataset compression, and this topic sound new to me. The definition of “subset degradation problem” is important in this domain. It will help the researchers to consider the problem. The experiments validate the effectiveness of the propose method.

**Strengths:**

1. The figures are beautiful and easy to understand.
2. The idea is novel. The process compression sound new to me since it is different from the model compression or dataset compression.
3. The “subset degradation problem” is practical. Although I have find similar pattern in experiments, it is good to see it is officially and properly presented.
4. The experiment results are promising. It save the computational cost by N times.

**Weaknesses:**

1. It's not clear what's the purpose of baseline B. It looks like the results are only compared to baseline A and C.
2. It's not clear why the freezing is used in MLS selection. If adaptive is good, why not just use adaptive method to choose the subset?
3. Will the additional loss bring extra computational cost?

**Questions:**

See weaknesses.

---

> ### Author Response · Authors · 2023-11-18
>
> Thank you for the positive feedback.
> We will address the listed questions one by one.
>
> > 1. It's not clear what's the purpose of baseline B. It looks like the results are only compared to baseline A and C.
>
> Before explaining the purpose of Baseline B, here is the recap of the all three Baselines. We will use examples of 10 images per class (IPC=10).
> - Baseline A:
> 	- The ultimate goal of this task, allowing **all subset** to have the same performance as directly condense to a dataset with specific IPC (Image Per Class).
> 	- **Approach**: conduct 10 separate condensations and obtain 10 different datasets.
> 	- The **purpose** is the provide an ''upper bound'' of the experimental result.
> - Baseline B:
> 	- **Approach**: Condense to 10 different datasets with IPC=1, construct large synthetic dataset using small IPC datasets.
> 	- **Purpose**: this approach **ensures** that the accuracy of the **smallest IPC** is preserved as opposed to preserving the accuracy of the largest IPC (Baseline C). The baseline B illustrates that combining many small datasets do not give comparable results on large IPC.
> - Baseline C:
> 	- **Approach**: traditional condensation approach.
> 	- **Purpose**: show the subset dagradation problem.
>
>  The main reason we introduce Baseline-B is that it has the **same storage** as Baseline-C while addressing the subset degradation problem (accuracy of IPC1 is preserved). However, as illustrated in the experimental results, Baseline-B introduces another problem: using many small synthetic images to construct a big datasets does not give comparable results. **That it, Baseline-B has good performance when IPC is small, but fails when IPC is large.
>  Baseline-C is the exact opposite, achieving good performance of large IPC, but fails at small one.**
>  Therefore, the proposed method effectively address the problem from both sides, achieving good performance when both IPC is small and large.
>
>  > 2. It's not clear why the freezing is used in MLS selection. If adaptive is good, why not just use adaptive method to choose the subset?
>
> Adaptive selection and Freezing serve for different purposes.
> - **adaptive selection**: it is not practical to incorporate the information of all IPC at the same time due to incurring more computations or slowing learning process. We use adaptive selection to find **one** IPC that is the most learnable. By adding information of **one** IPC at a time, the mentioned problem is addressed.
> - **freezing**: freezing aims to prevent the ''already learned" images from being overwritten by information of large IPC. By extracting a smaller subset, ideally speaking, the subset should not contain information of larger IPCs, otherwise the performance will be affected.
>
>
> | Calculate | Compare | Freeze | 1     | 2     | 3     | 4     | 5     | 6     | 7     | 8     | 9     | 10    | Avg.  |
> |-----------|---------|--------|-------|-------|-------|-------|-------|-------|-------|-------|-------|-------|-------|
> | -         | -       | -      | 27.49 | 38.50 | 45.29 | 50.85 | 53.60 | 57.98 | 60.99 | 63.60 | 65.71 | 67.50 | 53.15 |
> | ✓         | -       | -      | 49.35 | 48.27 | 50.00 | 52.30 | 54.20 | 58.29 | 60.90 | **63.63** | **65.90** | **67.63** | 57.08 |
> | ✓         | ✓       | -      | 40.12 | **54.91** | 56.02 | 56.12 | 56.18 | 59.74 | 61.68 | 63.41 | 65.56 | 67.01 | 58.08 |
> | ✓         | ✓       | ✓      | **49.55** | 53.75 | **56.39** | **59.33** | **58.13** | **60.62** | **62.06** | 63.59 | 65.25 | 66.79 | **59.55** |
>
> This is the Tab. 2 from our paper, it shows the difference of condensation with (row 4) and without (row 3) freezing.
> Row 3 delivers a poorer performance, especially IPC=1 when compared to Row 4.
> The issue is that performance of IPC1 is affected by large IPC since it is **not frozen** (keep receiving pixel updates according to larger IPC).
>
> > 3. Will the additional loss bring extra computational cost?
>
> Yes, we provide an example in `Section. 4.3 More Analysis: Redcued Training Time Needed`.
> The complete training of our method requires 30% more training time (**15.1 hrs** vs 11.5 hrs) on CIFAR-10 IPC10.
> However, to reach the average accuracy of the traditional approach, we only requires **0.6 hrs** compared to 11.5hrs (traditional approach).
> Fig. 4 in the paper presents a visualization of the comparison.

---

> > ### Comment · Reviewer_DoMB · 2023-11-22
> > **Official Comment by Reviewer DoMB**
> >
> > Thanks for the author's response. My concerns have been well addressed. I would keep my rating as accept.

---

### Official Review · Reviewer_6Dsd · 2023-10-29

**Soundness:** 4 excellent
**Presentation:** 3 good
**Contribution:** 4 excellent
**Rating:** 8
**Confidence:** 4

**Summary:**

This paper introduces the Multisize Dataset Condensation (MDC) method, aiming to address challenges associated with dataset condensation in on-device processing scenarios. The main innovation lies in the compression of multiple condensation processes into a single process to produce datasets of varying sizes. The authors combat the "subset degradation problem" with an "adaptive subset loss," ultimately enhancing the representation quality of condensed subsets. Experiments spanning various networks and datasets showcase the method's effectiveness.

**Strengths:**

Originality: This paper offers a unique approach to dataset condensation, aiming to cater to the specific needs of on-device scenarios. The proposal to compress N condensation processes into one is innovative.
Quality: The "adaptive subset loss" is a novel concept, targeting the "subset degradation problem." The method to select the Most Learnable Subset (MLS) is well-thought-out and complex.
Clarity: The paper is organized logically, and concepts are explained clearly. The use of terms like "adaptive subset loss" and "subset degradation problem" helps the reader understand the core issues being addressed.
Significance: The problem space being tackled (on-device training with dynamic computational resources) is relevant. Solving this issue can have substantial implications for real-world applications.

**Weaknesses:**

The paper explains three baselines for comparison. Compared to baseline A, the accuracy is not higher. Please explain the reason.
Is it possible to reach Baseline A's accuracies?
Equation 7 is not that clear. How to calculate the distance between the full dataset and subset?

**Questions:**

Please see weakness.

---

> ### Author Response · Authors · 2023-11-18
>
> Thank you for the positive rating. The main question provided has 3 small question, and we will address each of them accordingly.
>
> > 1. Compared to baseline A, the accuracy is not higher. Please explain the reason.
>
> Take IPC10 as an example, the Baseline-A is composed of 10 separate condensation processes and 10 different datasets.
> During the evaluation of the Baseline-A, we evaluate each synthetic dataset separately. Therefore, there is **NO INTERFERENCE** between datasets.
> Compare to the propose method, it requires 5x more storage.
> Our method, on the other hand, does have the inference between different subsets. For instance, IPC1 should not have information of large IPCs (IPC>1) during the evaluation of IPC1. However, during the evaluation of larger IPCs, we actually expect IPC1 to contain information of larger IPCs. Therefore, there is a conflict of interests and trade-offs should be made.
>
> > 2. Is it possible to reach Baseline A's accuracies?
>
> We believe it is possible to reach a similar average performance (a very close one), but we don't have the proof at this stage.
> One interesting observation is that: at larger IPCs, the performance of ImageNet outperforms Baseline-A. The below table is a taken from Tab. 1 (c) in our paper.
>
> | Dataset             |      |  10    | 15    | 20    |
> |---------------------|------|--------|-------|-------|
> | ImageNet-10         | A    |  **72.80** | 75.50 | 76.60 |
> |                     | B    |  63.60 | 62.73 | 64.13 |
> |                     | C    |  73.00 | 74.47 | 75.73 |
> |                     | Ours |  71.13 | **76.00** | **79.20** |
>
> The performance of IPC15 and IPC20 indeed outperforms the Baseline-A, and it exceeds by a noticeable margin.
> This gives us a sign that updating the subsets (smaller IPCs) may eventually help larger IPCs.
>
> > 3. Equation 7 is not that clear. How to calculate the distance between the full dataset and subset?
>
> The feature distance comparison is possible even if number of images are different.
> Since both real and synthetic images are fed into the network, the output feature only differs in the number of images.
> For example, if we sample 40 real images from full datasets and forward them, the output dimension is (40, 4, 4, 512), where 40 is the number of images, 4 is the feature size, and 512 is the number of channels. Lets say if we forward 20 synthetic images, the output will have shape like (20, 4, 4, 512), where the difference is on the first dimension.
> We then average the feature along the first dimension, we will then obtain two averaged features of shape (4, 4, 512). Therefore we can compare. A similar approach is used by Wang etal. [1].
>
> [1] K. Wang _et al._, “CAFE: Learning to Condense Dataset by Aligning Features,” in _CVPR_, 2022.

---

### Official Review · Reviewer_ZxGV · 2023-11-01

**Soundness:** 3 good
**Presentation:** 3 good
**Contribution:** 3 good
**Rating:** 8
**Confidence:** 4

**Summary:**

This paper proposes a method called Multisize Dataset Condensation (MDC) to compress multiple-size dataset condensation processes into a single process. The goal is to obtain a small synthetic dataset that is equally effective but much smaller in size. The authors introduce the concept of the Most Learnable Subset (MLS) and propose an adaptive subset loss to mitigate the "subset degradation problem" in traditional dataset condensation. The MDC method can reduce the condensing process and lower the storage consumption. The MDC achieves state-of-the-art performance on various models and datasets.

**Strengths:**

1. The proposed Multisize Dataset Condensation (MDC) method can effectively condense the N condensation processes into a single condensation process with lower storage and addresses the “subset degradation problem”.
2. The adaptive subset loss in the MDC method helps mitigate the “subset degradation problem” and improves the accuracy of the condensed dataset compared to the Baseline-C.
3. The concept of the rate of change of feature distance as a substitute for the computationally expensive “gradient distance” reduces computational overhead while capturing essential characteristics among subsets.

**Weaknesses:**

1. When the IPC (Inter-Process Communication) is small, there still exists a large accuracy gap between the proposed model and Baseline-A as shown in Figure 2 and Table 1.
2. The impact of the calculation interval (∆t) on the performance of the MDC method needs to be further analyzed to determine the optimal interval size.

**Questions:**

1. Can you provide the computational resource consumption and algorithmic complexity compared to Baseline-A, B, C, and other SOTA methods? It can help authors better understand the effects of algorithms in devices with limited computational resources.
2. Can you provide the values of hyperparameters such as λ and η in Formula 2?
3. The section on Visualization of MLS is currently difficult to understand. It would be helpful to provide more detailed and accessible explanations to ensure a clear understanding for readers.

---

> ### Author Response · Authors · 2023-11-18
>
> Thank you so much for bringing up the questions and concerns. We try to address them one by one.
>
> > 1. When the IPC (Inter-Process Communication) is small, there still exists a large accuracy gap between the proposed model and Baseline-A as shown in Figure 2 and Table 1.
>
> The IPC we use stands for Image Per Class, indicating how many images are for each class.
> The Baseline-A uses much more resources compared to the proposed method. It is the **ideal setting** if the resources being used is the same as our method.
> To the best of our knowledge, there is no method could achieve Baseline-A using the given resources (i.e., one condensation process with one dataset).
>
> > 2. The impact of the calculation interval (∆t) on the performance of the MDC method needs to be further analyzed to determine the optimal interval size.
>
> Optimizing the calculate interval $\Delta t$ will further improve the performance, and we admit finding the optimal one will not be easy, especially this hyper-parameter can vary from dataset to dataset.
> We will leave this problem for future exploration.
> Nevertheless, Tab. 7 shows that the choice of $\Delta t$ fall a wide range, and each of them brings a considerable improvement.
>
> > 3. Can you provide the computational resource consumption and algorithmic complexity compared to Baseline-A, B, C, and other SOTA methods? It can help authors better understand the effects of algorithms in devices with limited computational resources.
>
> Tab. 2 (b) compares the resources for different baselines. (A simplified version is shown below.)
>
> |      | Condense | Storage         |
> | ---- | -------- | --------------- |
> | A    | N        | 1 + 2 + ... + N |
> | B    | N        | N               |
> | C    | 1        | N               |
> | Ours | 1        | N               |
>
> Take CIFAR-10 as an example, condensing using IDC takes roughly 11 hours on average for IPC1-10. Note the time used to condense to different IPCs is different, but will not be extremely different, e.g., IPC1 ~ 10 hrs, IPC10 ~ 12 hrs.
> Here are the resources needed for each baseline:
> - Baseline-A: 110 hours + 550 images (storage)
> - Baseline-B: ~100 hours + 100 images
> - Baseline-C: 11.5 hours + 100 images
> - Ours (+0.0%): 0.6 hours + 100 images
> - Ours (+6.4%): 15.1 hours + 100 images
>
> In conclusion, our method incurs 30% more training cost to attain 6.4% increase of average accuracy. To attain the same accuracy as Baseline-C, **0.6** hrs is enough. The visualization is presented in Fig. 4 in the paper.
>
> **Comparing to other SOTA.** We did not conduct experiments on other SOTA except for IDC-based method, but the expected additional cost is 30% - 50% (capped at 100%). The reason is that the selected IPCs are usually not large, incurring small additional cost (as shown in Fig. 5).
>
> > 4. Can you provide the values of hyperparameters such as λ and η in Formula 2?
>
> The $\lambda$ and $\eta$ are learning rates using in dataset distillation frameworks, and we did not modify this.
> We follow the same setting used in IDC [1] (Details can be found in Appendix C.1 in [1]).
>
> [1] J.-H. Kim _et al._, “Dataset Condensation via Efficient Synthetic-Data Parameterization,” in _ICML_, 2022.
>
> > 5. The section on Visualization of MLS is currently difficult to understand. It would be helpful to provide more detailed and accessible explanations to ensure a clear understanding for readers.
>
> Thank you for the feedback, we will try to adjust the figure for a clearer presentation.

---

> > ### Comment · Reviewer_ZxGV · 2023-11-22
> >
> > Thanks for the author's response. My concerns have been essentially addressed.

---

### Official Review · Reviewer_CWAb · 2023-11-04

**Soundness:** 3 good
**Presentation:** 3 good
**Contribution:** 3 good
**Rating:** 6
**Confidence:** 3

**Summary:**

This paper introduces the Multisize Dataset Condensation problem that can derive multiple subsets from the condensed images for supporting on-device scenarios. The authors identify “subset degradation problem” where the performance of a subset from condensed images is lower than directly condensing the full dataset to the target size. Subsequently, the authors propose “adaptive subset loss” where the most learnable subset is selected to update the subset, to alleviate the “subset degradation problem” for all subsets. Experimental results demonstrate that MDC works well for various datasets.

**Strengths:**

The paper presents a solution for DC named Multisize Dataset Condensation which is crucial for on-device scenarios. The proposed method outperforms baseline C significantly.

**Weaknesses:**

1. The synthetic samples within the subset seem to be fixed, which may not reflect “Multisize Dataset Condensation” correctly.

**Questions:**

I have several questions:
1. In Fig 2c, for baseline C, how to select subsets to calculate accuracy? Is it random? Let’s assume we have a subset of 2 images. Do we select 2 images from the condensed data randomly?
2. In basic condensation training (Sec 4.1), for each initialization the network is trained for 100 epochs. Is it the inner loop E?

---

> ### Author Response · Authors · 2023-11-18
>
> Thank you for your comments. We will response point by point.
>
> > 1. The synthetic samples within the subset seem to be fixed, which may not reflect “Multisize Dataset Condensation” correctly.
>
> The keyword (**multisize**) in our title means the condensed dataset can be treated as a multi-sized dataset.
> That is, the dataset of $IPC=N$ can be used as dataset of $IPC={1, 2, ..., N}$ without requiring additional condensation process.
> This is currently unattainable using traditional approach, since traditional one condenses a synthetic dataset to a **specific IPC**. For example, dataset with $IPC=N$ will perform poorly on smaller $IPC$ ones, and we take the rights to name the phenomenon "Subset Degradation Problem". Our method addresses the problem, turning the dataset available for multiple size of IPCs.
>
> > 2. In Fig 2c, for baseline C, how to select subsets to calculate accuracy? Is it random? Let’s assume we have a subset of 2 images. Do we select 2 images from the condensed data randomly?
>
> For baseline C, all subsets are selected using the $N$ images. For example, if there are $10$ images in the class, we take the first image as IPC=1, and first two images as IPC=2.
>
> To address the concern, here is the result of randomly sampled images (see below table). We do not see an obvious advantage with one method over the other.
>
> |          | 1        | 2        | 3        | 4        | 5        | 6    | 7    | 8        | 9        | 10    | avg  |
> | -------- | -------- | -------- | -------- | -------- | -------- | ---- | ---- | -------- | -------- | ----- | ---- |
> | First N  | 27.5     | 38.5     | **45.3** | 50.9     | **53.6** | 58.0 | 61.0 | 63.6     | **65.7** | 67.50 | 53.2 |
> | Random N | **28.4** | **39.3** | 44.3     | **51.1** | 53.2     | 58.0 | 61.0 | **63.8** | 65.6     | 67.50 | 53.2 |
>
> > 3. In basic condensation training (Sec 4.1), for each initialization the network is trained for 100 epochs. Is it the inner loop E?
>
> Yes. The 100 epoch is exactly the inner loop E depicted in Fig. 3.
>
> We hope the response addresses your concerns.

---

### Meta-Review · Area_Chair_1WwH · 2023-12-08

**Metareview:**

The paper proposes a novel dataset distillation method to compressing multiple condensation processes into a single condensation process, thus achieving a flexible size of the condensed dataset. All reviewers have unanimously agreed on the acceptance of this paper. The authors have successfully and convincingly addressed each point of feedback including the meaning of three baselines and extra computational cost caused. I believe this new perspective could inspire further research in the field of data distillation. Therefore, I strongly recommend the acceptance of this paper.

**Justification For Why Not Higher Score:**

N/A

**Justification For Why Not Lower Score:**

Considering the robust reviews and high scores (8, 8, 8, and 6) for submission 801, I advocate for its oral presentation at ICLR due to its potential impact and methodological innovations. Key points include:

1. Review 6Dsd notes the paper's "unique approach to dataset condensation," suggesting a new paradigm for data efficiency.

2. As mentioned by Review ZxGV, the "adaptive subset loss" addresses the "subset degradation problem," potentially revolutionizing how we approach dataset optimization and storage.

3. The method demonstrates substantial computational savings and improved performance, with Review DoMB highlighting its novel approach and practicality.

It has implications for enhancing dataset compression techniques across fields including dataset pruning and condensation.It could influence a range of methods, from on-device learning to real-time data processing. So I would recommend to Oral.

---

### Decision · Program_Chairs · 2024-01-16

Accept (oral)